# Murine hematopoietic stem cell activity is derived from pre-circulation embryos but not yolk sacs

Miguel Ganuza [1], Ashley Chabot[1], Xing Tang[1], Wenjian Bi[2], Sivaraman Natarajan [3], Robert Carter[3], Charles Gawad[3], Guolian Kang[2], Yong Cheng[1] & Shannon McKinney-Freeman [1]

The embryonic site of definitive hematopoietic stem cell (dHSC) origination has been debated for decades. Although an intra-embryonic origin is well supported, the yolk sac (YS) contribution to adult hematopoiesis remains controversial. The same developmental origin makes it difficult to identify specific markers that discern between an intraembryonic versus YS-origin using a lineage trace approach. Additionally, the highly migratory nature of blood cells and the inability of pre-circulatory embryonic cells (i.e., 5–7 somite pairs (sp)) to robustly engraft in transplantation, even after culture, has precluded scientists from properly answering these questions. Here we report robust, multi-lineage and serially transplantable dHSC activity from cultured 2–7sp murine embryonic explants (Em-Ex). dHSC are undetectable in 2–7sp YS explants. Additionally, the engraftment from Em-Ex is confined to an emerging $CD31^+CD45^+c\text{-}Kit^+CD41^-$ population. In sum, our work supports a model in which the embryo, not the YS, is the major source of lifelong definitive hematopoiesis.

[1] Department of Hematology, St. Jude Children's Research Hospital, Memphis, TN 38105, USA. [2] Department of Biostatistics, St. Jude Children's Research Hospital, Memphis, TN 38105, USA. [3] Department of Oncology, St. Jude Children's Research Hospital, Memphis, TN 38105, USA. Correspondence and requests for materials should be addressed to S.M-F. (email: Shannon.McKinney-Freeman@stjude.org)

The embryonic origin of cells that sustain lifelong mammalian hematopoiesis and blood production has long been debated. Resolving this debate is complicated by the emergence of sequential waves of blood cells at distinct sites within the embryo:[1] blood-islands composed of primitive nucleated erythrocytes first appear at E7-E7.5 in the YS. Definitive erythroid-myeloid precursors also emerge from the YS at E8.5. Finally, around E10.5-E11.5, the first definitive HSC (dHSC) capable of reconstituting the hematopoietic system of adult recipients using existing assays are detected and presumably these precursors support lifelong blood production[2,3]. The site of origin of these dHSC has been contentious[2–16]. An intra-embryonic origin, concentrated around the para-aortic splanchnopleura (PSp)-derived aorta-gonad-mesonephros region (AGM), is currently the favored model. In contrast, the contribution of YS to the dHSC compartment is controversial[1]. Early work implicated the YS blood islands as a source of both primitive-erythroblasts and dHSC;[1,4–6,8,15] however later work challenged this hypothesis. In particular, Dieterlen-Lievre and colleagues demonstrated an intra-embryonic origin for definitive hematopoiesis in vertebrates using quail-chick chimeras[7,16]. Recent work has formally demonstrated in chicken the presence of bona fide dHSC originating from the embryo aortas but not from the YS, allantois or head[17]. An intra-embryonic origin for dHSC in mammals was later supported by studies showing that the first dHSC capable of reconstituting adult recipients are detected in the PSp/AGM region[2,3]. Despite these findings, the potential contribution of YS to lifelong hematopoiesis has not been completely excluded[13,14,18,19].

YS-derived and AGM-derived hematopoietic progenitors both arise from hemogenic endothelial (HE) precursors that are mesodermal in origin[14,20–25]. Very few markers have been identified that could potentially distinguish between AGM and YS hematopoietic precursors. The highly migratory nature of blood cells in circulating embryos and the inability of cells isolated from pre-circulation embryos to robustly engraft in transplantation assays, even after ex vivo culture, has precluded definitively addressing if the YS hemogenic endothelium (YS-HE) contributes to lifelong hematopoiesis and the adult dHSC pool[12,26]. PSp tissue from pre-circulation embryos generated long-term multi-lineage engraftment while YS did not, but reconstitution was extremely low (1–5%) in these experiments, raising concerns that lower activity present in the YS would have been very difficult to detect[12]. Furthermore, PSp-derived reconstitution was only observed in severely immunocompromised recipient mice (i.e., Rag2γc$^{-/-}$)[12]. Indeed, it has recently been suggested that the YS may be a major embryonic source of dHSC[14]. Lineage tracing studies exploiting the high expression of LYVE1 (lymphatic vessel endothelial hyaluronan receptor-1) in the YS and vitelline-endothelium concluded that >40% of adult blood may ultimately derive from these sites in mice[14].

Here, we present a platform that supports the ex vivo development of robust dHSC activity from pre-circulation embryos, allowing us to rigorously interrogate the dHSC-forming potential of both the early embryo and YS. We find that cultured pre-circulatory Em-Ex, but not YS explants (YS-Ex), yield robust dHSC activity. Importantly, this activity in cultured Em-Ex was restricted to an emerging CD31$^+$CD45$^+$c-Kit$^+$CD41$^-$ population that also develops in cultured YS-Ex. Additionally, in pre-circulation embryos, we identify LYVE1$^+$CD31$^+$ aortic endothelial cells, confirming that LYVE1 expression is found outside the YS and vitelline HE at this early stage of development[14]. We further demonstrate that pre-circulatory Em-Ex-derived LYVE1$^+$ precursors yield robust dHSC activity, indicating that LYVE1 constitutes an early marker of intraembryonic definitive

hematopoiesis. In sum, our work strongly supports a model in which the YS is not a major source of lifelong definitive hematopoiesis.

## Results

**Robust dHSC activity generation from pre-circulatory embryos.** Blood circulation is established around 5–7sp (≈E8.5). The first heartbeat is detected at 5sp, a consistent blood flow begins around 7sp and fully functional circulation is established after E10, as evidenced by the uneven distribution of erythroblasts within the embryo vasculature prior to 35sp[27,28]. Although dHSC activity has previously been observed from cultured pre-circulation embryos, this activity was low, and only observed after transplantation into Rag2γc$^{-/-}$ immunocompromised recipients[12]. Here, we sought to determine if it might be possible to coax robust dHSC activity from E8-E8.5 concepti.

We collected CD45.2$^+$ embryos at E8.0-E8.5 and staged them by number of somite pairs. Concepti were organized into two groups: 2–3sp (pre-circulation) and 5–7sp (peri-circulation). Embryos were separated from the YS and cultured at the air-liquid interface in the presence of SCF, FLT3L, and IL-3, as previously described[29–31]. We hypothesized that an extended culture period might be required for maturation of dHSC from these early embryos (Fig. 1a). Hence, embryos were cultured for 9–10 days. Cultured Em-Ex were then recovered, dissociated and transplanted into lethally irradiated CD45.1$^+$CD45.2$^+$ recipients along with $2 \times 10^5$ CD45.1$^+$ whole bone marrow (WBM) cells for radio-protection (Fig. 1a). Remarkably, both 2–3sp and 5–7sp cultured embryos generated robust dHSC activity. 5–7sp embryos engrafted 9/11 (82%) recipients while 2–3sp embryos engrafted 5/7 (71%) recipients (Fig. 1b). Globally, >15% of recipient peripheral blood (PB) was CD45.2$^+$ in 7/18 total recipients at 16 weeks post-transplant. Four recipients displayed >50% CD45.2$^+$ PB reconstitution (Fig. 1b). Importantly, most engrafted recipients (11/14) displayed multi-lineage engraftment at 16 weeks post-transplant (Fig. 1b). To further test if Em-Ex developed robust dHSC activity, we performed secondary limiting dilution transplants. $0.8 \times 10^6$ ($n = 9$), $1 \times 10^6$ ($n = 6$), $2 \times 10^6$ ($n = 8$) or $5 \times 10^6$ ($n = 8$) WBM cells from three independent primary recipients of 5–7sp Em-Ex were transplanted into lethally irradiated CD45.2$^+$CD45.1$^+$ recipients (Fig. 2a). At the three highest doses of WBM, 100% of secondary recipients displayed significant Em-Ex-derived CD45.2$^+$ chimerism (1–92%) at 16 weeks post-transplant (Fig. 2a). CD45.2$^+$ reconstitution was >25% in 11/31 secondary recipients and >90% in two recipients. Multiple independent secondary recipients also yielded robust long-term Em-Ex-derived CD45.2$^+$ chimerism after transplantation into lethally irradiated CD45.1$^+$CD45.2$^+$ tertiary recipients, although interestingly this reconstitution displayed dramatic lineage skewing (Fig. 2b). The robust hematopoietic reconstitution of primary, secondary, and tertiary recipients by cultured 5–7sp Em-Ex is indicative of self-renewing dHSC and long-term repopulating cells, confirming that our explant culture system supports the emergence and maturation of dHSC from embryos isolated at the onset of blood circulation.

**Pre and peri-circulation YS do not produce dHSC ex vivo.** Dissected CD45.2$^+$ YS from pre-circulation and peri-circulation concepti (2–3sp and 5–7sp) were cultured as explants (YS-Ex) in parallel and under the same conditions as Em-Ex. None of the cultured pre-circulation or peri-circulation CD45.2$^+$ YS-Ex yielded detectable dHSC activity when transplanted into lethally irradiated CD45.1$^+$CD45.2$^+$ recipients along with $2 \times 10^5$ WBM CD45.1$^+$ cells (0/5 and 0/11 recipients, respectively, Fig. 1b). It is

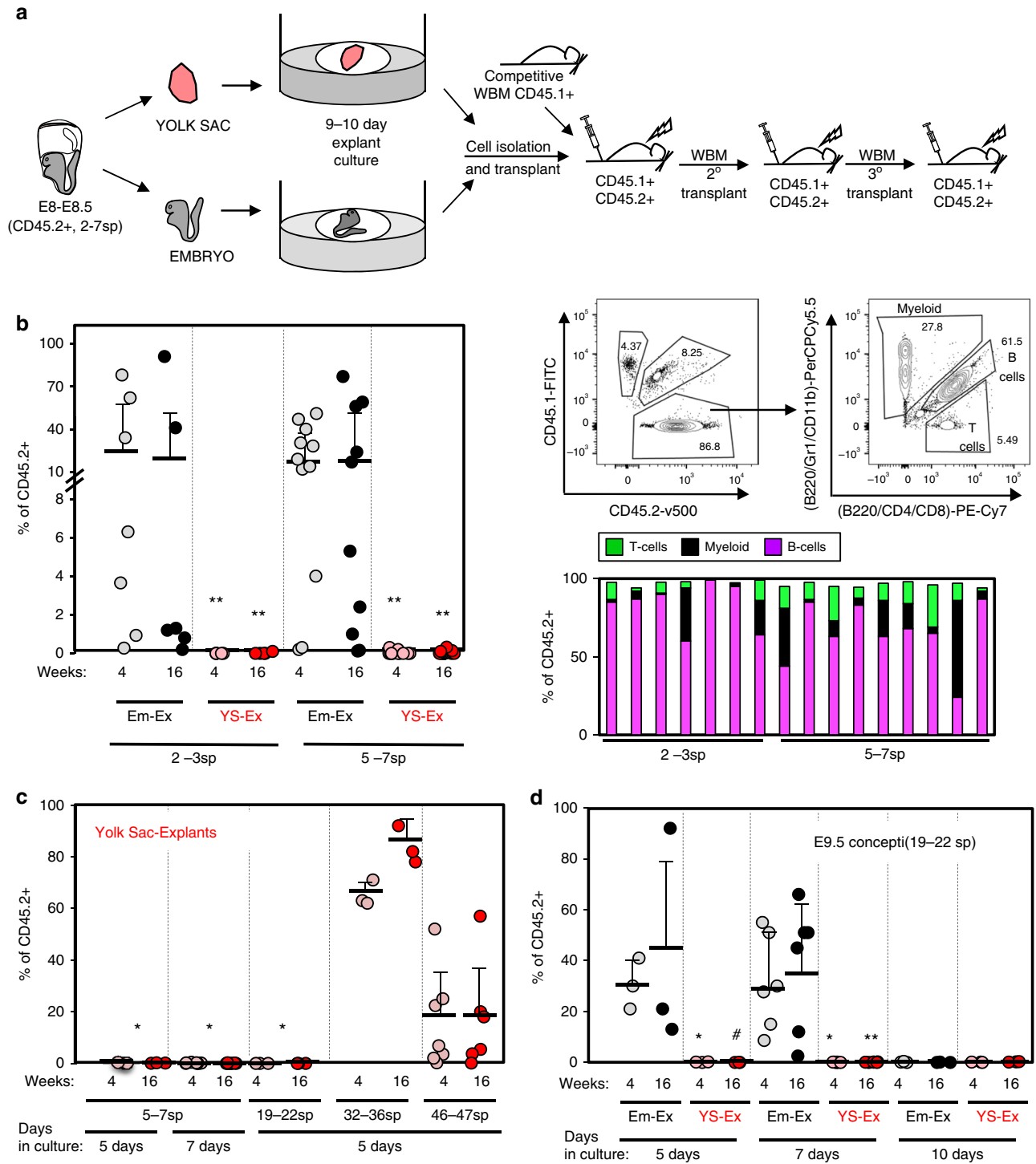

possible that YS hemogenic precursors undergo distinct temporal maturation compared to intra-embryonic hemogenic endothelium. Indeed, YS-Ex displayed substantially greater CFU-GEMM (i.e., colony forming unit granulocyte, erythrocyte, monocyte, megakaryocytic) potential at day 5 versus day 10 of ex vivo culture (Supplementary Fig. 1A). Thus, 5–7sp YS-Ex were cultured for five or seven days prior to dissociation and transplanted into lethally irradiated recipients. These cultures also failed to produce detectable dHSC activity (0/8 for five-day YS-Ex cultures and 0/3 recipients for seven-day YS-Ex, Fig. 1c).

These data indicate that the only precursors capable of maturing into functional dHSC under these conditions are found

in the embryo at this early stage of development. Additionally, we tested the ability of YS isolated from later developmental time-points (e.g., E9.5 (19–22 sp, $n = 3$), E10.5 (32–36 sp, $n = 3$) and E11.5 (46–47sp, $n = 7$)) to engraft lethally irradiated recipients after being cultured ex vivo for five days prior to transplantation. 100 and 71% of CD45.1+CD45.2+ recipients transplanted with CD45.2+ E10.5 and E11.5 cultured YS-explants showed robust CD45.2+ YS-derived engraftment, respectively (Fig. 1c). In contrast, CD45.1+CD45.2+ recipients transplanted with CD45.2+ E9.5 YS-derived explants displayed no evidence of CD45.2+ engraftment (Fig. 1c). To test if E9.5 hematopoietic precursors are capable of maturing into dHSC under our ex vivo

**Fig. 1** Pre-circulation embryos but not YS yield robust dHSC activity. **a** Experimental schematic. Em and YS were dissected from E8-E8.5 (2–7sp) concepti and cultured as explants at the air-liquid interface for 9–10 days. Explants were then harvested, dissociated and transplanted at ≥four embryo equivalent (e. e.)/recipient. **b** Left panel: CD45.2[+] (i.e., 2–3sp Em-Ex ($n = 7$), 5–7sp Em-Ex ($n = 11$), 2–3sp YS-Ex ($n = 5$) or 5–7sp YS-Ex ($n = 11$)) contribution to PB of primary recipients. Data pooled from 16 independent experiments. Generally, tissues from one independent litter were transplanted per experiment. For 2–3sp explants, all YS-Ex ($n = 5$) were transplanted in parallel with Em-Ex isolated from the same concepti. For 5–7sp explants, all YS-Ex ($n = 11$) were transplanted in parallel with at least one Em-Ex isolated from the same concepti as a positive control for engraftment (**$p < 0.01$, eWrs-test). Right panel: Frequency of myeloid cells, T cells and B cells in CD45.2[+] PB and representative flow cytometry plots of primary recipients of 2–3sp and 5–7sp Em-Ex 16 weeks post-transplant. Each column represents an independent recipient. For the 5–7sp Em-Ex, only the engrafted mice are shown ($n = 9$). **c** CD45.2[+] YS were isolated from E8.5 (5–7sp, $n = 8$ recipients), E9.5 (19–22sp, $n = 3$ recipients), E10.5 (32–36sp, $n = 3$ recipients) or E11.5 (46–47sp, $n = 7$ recipients) concepti, cultured as explants for five or seven days and then transplanted at eight ee/recipient. Data pooled from five independent experiments. (*$p < 0.05$, eWrs-test). **d** YS-Ex and Em-Ex were isolated from CD45.2[+] E9.5 (19–22sp) concepti and cultured under the same conditions as in **b** for five ($n = 3$ recipients), seven ($n = 6$ recipients) or 10 days ($n = 3$ recipients) before transplantation into lethally irradiated recipients at eight ee/recipient. Statistical differences between Em-Ex and YS-Ex are indicated (**$p < 0.01$, *$p < 0.05$, #$p < 0.1$, two-sample $t$-test and eWrs-test). **b–d** %CD45.2[+] PB (i.e., YS-Ex or Em-Ex-derived) was examined in recipients four and 16 weeks post-transplantation. Each circle represents an independent recipient. For all panels, gray or pink and black or red circles indicate four and 16 weeks post-transplant, respectively. Means and standard deviations are shown. Source data are provided as a Source Data file

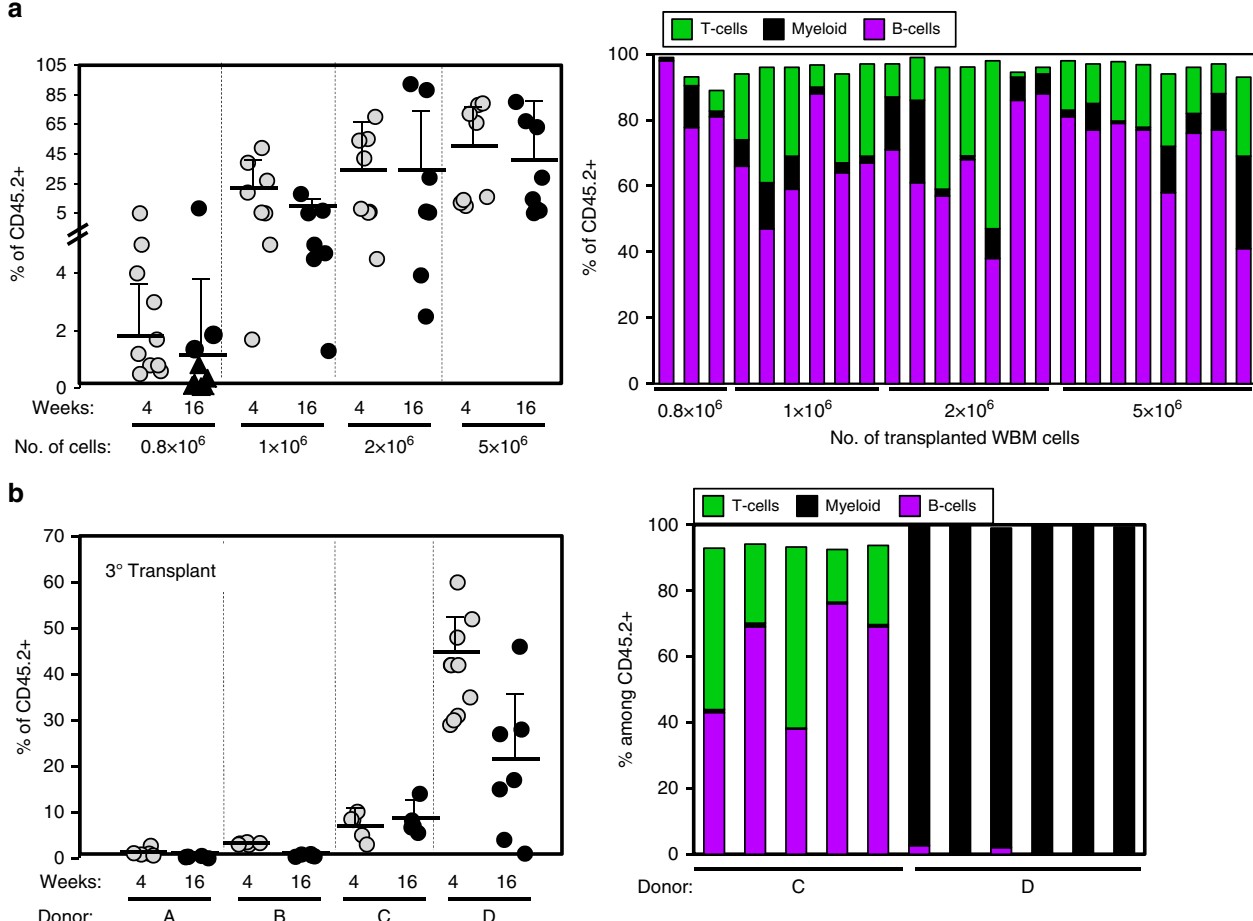

**Fig. 2** Pre-circulation Em-Ex display serially transplantable multi-lineage potential. **a** Left panel: Total WBM was isolated from three primary recipients of CD45.2[+] 5–7sp Em-Ex-derived cells and transplanted separately into three secondary cohorts of lethally irradiated CD45.1[+]CD45.2[+] recipients. Mice received $0.8 \times 10^6$ ($n = 9$), $1 \times 10^6$ ($n = 7$), $2 \times 10^6$ ($n = 8$) or $5 \times 10^6$ ($n = 8$) total WBM cells. Data pooled from three independent experiments is shown. Non-engrafted mice are depicted as triangles. Means are shown. Error bars denote standard deviation. Right panel: Frequency of myeloid cells, T cells and B cells in CD45.2[+] PB of secondary recipients of 5–7sp Em-Ex 16 weeks post-transplant. **b** $5 \times 10^6$ WBM cells from secondary recipients of 5–7sp Em-Ex were transplanted into CD45.1[+]CD45.2[+] mice along with $2 \times 10^5$ CD45.1[+] WBM cells. Bone marrow from $n = 4$ independent secondary recipients were transplanted into independent cohorts of tertiary recipients. Secondary donors A, B, and C derived from the same initial primary recipient. Secondary donor D derived from an independent primary recipient. Left panel: %CD45.2[+] tertiary recipient PB cells. Four (white circles) and 16 (black circles) weeks post-transplant. Donors A, B, and C ($n = 5$ recipients); Donor D ($n = 10$ recipients). Each circle indicates an independent tertiary recipient. Means and standard deviations are depicted. Right panel: Frequency of myeloid cells, T cells and B cells in CD45.2[+] PB of tertiary recipients 16 weeks post-transplant. Each column corresponds to an independent recipient. Source data are provided as a Source Data file

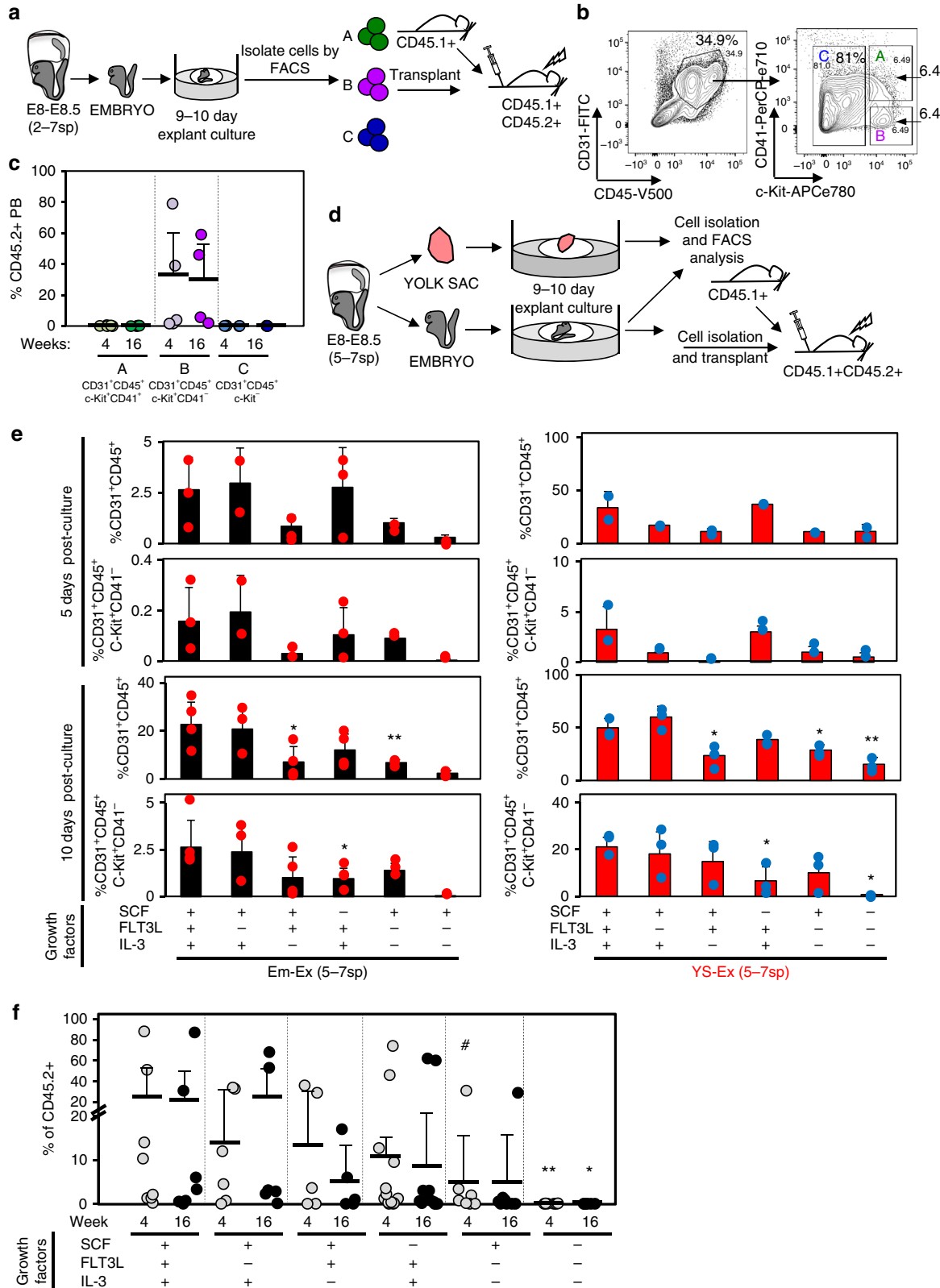

culture conditions, embryonic caudal halves and YS isolated from CD45.2$^+$ E9.5 concepti were cultured for five, seven or 10 days and then transplanted into lethally irradiated CD45.1$^+$CD45.2$^+$ recipients (Fig. 1d). 100% of recipients transplanted with embryonic caudal halves cultured for five or seven days (3/3 and 6/6, respectively, Fig. 1d) displayed robust CD45.2$^+$ dHSC activity, confirming that post-circulation E9.5 dHSC can mature

under these culture conditions. However, all E9.5 YS-derived explants failed to generate detectable dHSC in these experiments (Fig. 1d).

In sum, these findings support a model where the precursors in pre-circulation and peri-circulation concepti capable of maturing into functional dHSC under these conditions are located in the embryo.

**Fig. 3** HSC potential is restricted to CD31+CD45+c-Kit+CD41− embryo explant cells and requires SCF and IL-3. **a** Experimental schematic. Embryos isolated from E8.5 (5–7sp) concepti were cultured as in Fig. 1b, dissociated and fractionated by FACS based on CD31, CD45, and c-Kit cell surface expression and then transplanted at four ee/recipient into lethally irradiated CD45.1 + CD45.2 + recipients along with 2 × 10^5 CD45.1+ WBM cells. **b** Representative flow cytometry plots denoting sorted and transplanted populations (A: CD31+CD45+c-Kit+CD41+; B: CD31+CD45+ c-Kit+CD41−; C: CD31+CD45+c-Kit−). **c** %CD45.2+ PB of recipients at four (light-colored circles) and 16 weeks (dark-colored circles) post-transplant. Each circle represents an individual recipient. Five independent experiments are depicted. In each experiment all sorted cells for each population were transplanted into one separate recipient. Engraftment was only detected upon transplantation of population B. **d** Experimental schematic. E8.5 (5–7sp) YS-Ex and Em-Ex were cultured at the air-liquid interface for 9–10 days in different combinations of SCF, FLT3L and IL-3 and either analyzed by flow cytometry or transplanted to test for dHSC. **e** Cultured explants were analyzed by flow cytometry for CD31+CD45+ and CD31+CD45+c-Kit+CD41−cells after 5 (n = 2, with the exception of Em-Ex SCF + IL-3 + FLT3L +, Em-Ex SCF + IL-3 + and Em-Ex IL-3 + FLT3L + where n = 3) or 10 days of culture (n = 3, with the exception of Em-Ex SCF + IL-3 + FLT3L +, Em-Ex SCF + IL-3 + and Em-Ex IL-3 + FLT3L + where n = 4). Each circle represents an individual culture. Bars stand for average. Results from four independent experiments are shown. Left panel: Em-Ex-explants. Right panel: YS-Ex. Statistical differences with the SCF + Flt3l + IL-3 + control are indicated: (*p < 0.05 and **p < 0.01, two-sample t-test). **f** CD45.2+ Em-Ex were transplanted into lethally irradiated CD45.1 +CD45.2 + recipients at four ee/recipient along with 2 × 10^5 CD45.1+ WBM cells. %CD45.2+ PB four (gray circles) and 16 weeks (black circles) post-transplant is shown. Each circle represents an individual recipient. Cumulative data from 13 independent experiments. The majority of the analyzed conditions yielded an engraftment that was marginally statistically higher than the No-Growth Factor condition (**p < 0.01, *p < 0.05, #p < 0.1; eWrs-test and two-sample t-test). Horizontal bars indicate averages. Error bars denote standard deviation. Source data are provided as a Source Data file

**SCF and IL-3 induce CD31+CD45+c-Kit+CD41− dHSC emergence**. We next sought to phenotypically characterize functional dHSC that emerge during the ex vivo culture of peri-circulation Em-Ex. We took advantage of cell surface markers to isolate distinct populations from CD45.2+ 5–7sp Em-Ex for transplantation into lethally irradiated CD45.1+CD45.2+ recipients (Fig. 3a). Previous work showed that dHSC activity in E10.5-E11 Em-Ex cultured under similar conditions is contained in the CD31+CD45+ fraction[29,31]. This population was apparent in the Em-Ex cultures (Fig. 3b). CD41 and c-Kit are also expressed by nascent hematopoietic populations in early embryos[30–38]. Distinct populations expressing both CD41 and c-Kit were clear within CD31+CD45+ Em-Ex cells (Fig. 3b, Fig. 4b and Supplementary Fig. 1B). We therefore isolated CD31+CD45+c-Kit+CD41+, CD31+CD45+c-Kit+CD41−and CD31+CD45+c-Kit− cells by FACS from cultured 5–7sp Em-Ex and transplanted them into lethally irradiated CD45.1+CD45.2+ recipients (n = 5, Fig. 3a–c). All dHSC activity was restricted to CD31+CD45+c-Kit+CD41-cells (Fig. 3c). Interestingly, this same population was also apparent in 5–7sp YS cultures (Fig. 4b and Supplementary Fig. 1B).

We next tested whether the emergence of the CD31+CD45+c-Kit+CD41− engrafting population was selectively dependent on any of the three growth factors added to these cultures (i.e., SCF, IL-3, FLT3L). Embryos and YS isolated from 5–7sp concepti were cultured in the presence of different combinations of growth factors and assessed by flow cytometry for the emergence of CD31+CD45+ and CD31+CD45+c-Kit+CD41− cells (Fig. 3d, e). CD45.2+ Em-Ex cultures were also transplanted into lethally irradiated CD45.1+CD45.2+ recipients to test for dHSC activity (Fig. 3d, f). Removal of FLT3L had no effect on the frequency of CD31+CD45+ or CD31+CD45+c-Kit+CD41− cells in YS-Ex or Em-Ex cultures (Fig. 3e). Moreover, removal of FLT3L did not affect the emergence of dHSC in Em-Ex cultures: the contribution of CD45.2+ cells to the PB of transplant recipients was 18% in the presence of all three factors and 21% in the absence of FLT3L (Fig. 3e, f and Supplementary Fig. 1B). In contrast, removal of IL-3 or SCF reduced the frequency of CD31+CD45+ or CD31+CD45+c-Kit+CD41− cells, respectively, in YS-Ex and Em-Ex cultures, which correlated with a loss of dHSC activity from CD45.2+ Em-Ex cultures (Fig. 3e, f and Supplementary Fig. 1B). The average CD45.2+ hematopoietic reconstitution in these recipients dropped to about 5 and 10% in the absence of IL-3 or SCF, respectively (Fig. 3f). Em-Ex cultured in the absence of both FLT3L and IL-3 also displayed reduced hematopoietic engraftment potential (p < 0.1, exact Wilcoxon rank sum test (eWrs-test)) (Fig. 3f). Em-Ex cultured in the absence of all three cytokines displayed dramatic

reductions in the frequency of CD31+CD45+ and CD31+CD45+c-Kit+CD41− cells and lacked detectable dHSC activity (Fig. 3e, f and Supplementary Fig. 1B). In sum, administration of both SCF and IL-3 is required for the efficient development of CD31+CD45+ c-Kit+CD41− cells from pre-circulation concepti during YS-Ex and Em-Ex culture and dHSC potential in Em-Ex cultures. These results are consistent with the observation that SCF and IL-3 can both promote the in vitro derivation of bona fide HSC from other cell sources[29–31,39–41].

**An HSC-like population is generated by embryo explants**. YS and embryos are separated and cultured before the onset of the circulation in our experimental platform. Thus, this represents an opportunity to selectively study embryo versus YS-derived hematopoiesis under controlled conditions that allow for the maturation of dHSC from early embryonic precursors. Indeed, novel cell surface markers of dHSC-derived from early embryos could be useful reporters in efforts to isolate dHSC from pluripotent stem cells or other sources. Although CD31+CD45+c-Kit+CD41−cells develop in both YS-Ex and Em-Ex cultures, cultured YS-Ex-CD31+CD45+c-Kit+CD41− cells do not harbor dHSC activity (Figs. 1b, 3b, 3f, Supplementary Fig. 1B). Therefore, we decided to interrogate the gene expression profiles of 5–7sp YS-Ex-CD31+CD45+c-Kit+CD41− cells and Em-Ex-CD31+CD45+c-Kit+CD41− cells on day 10 post-culture for molecularly distinct cell populations that might explain the absence of dHSC activity in YS explants by single cell RNA sequencing (scRNAseq) (Fig. 4).

First, we determined the frequency of the engrafting population within cultured Em-Ex. Limiting dilution transplantation revealed 0.83 repopulating units (RUs) per Em-Ex (95% C.I. = (0.48, 1.42); results fit the LDA model: p-value = 0.717, t-test) (Fig. 4a). Next, 49 concepti were dissected and cultured as separate YS-Ex and Em-Ex for scRNAseq. In this experiment we are interrogating approximately 40.67 RUs in the Em-Ex sample (i.e.,: 49Em-Ex*0.83RUs).

To identify Em-Ex-derived populations of putative biological interest, the resulting scRNAseq data was projected as tSNE1 Vs tSNE2. A population specific to the Em-Ex sample was apparent (X-cells, Fig. 4c). It was comprised of 623 cells (17.8% of the Em-Ex population). To interrogate the developmental state of this population, we compared its expression profile to Zhou and colleagues scRNAseq dataset of embryonic hematopoiesis (Fig. 4d)[42]. Em-Ex cells were most similar transcriptionally to E11-T2-pre-HSC (CD31+CD45+cKit+CD41^low, 60% of the X-

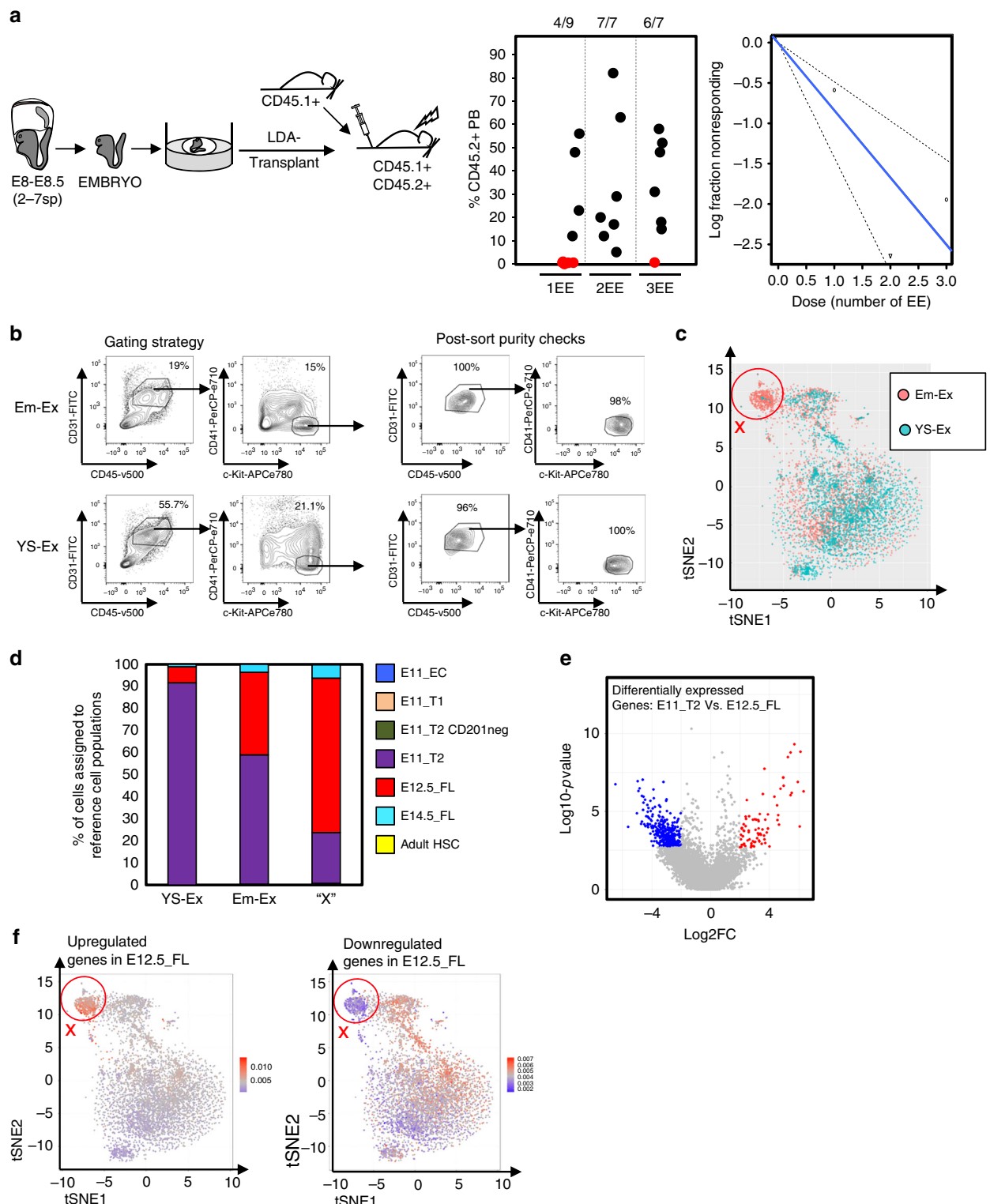

cells were assigned to this group) or E12.5 Fetal Liver (FL) HSC (Lineage−Sca-1+Mac-1lowCD201+, 37% of the X-cells were assigned to this group) and thus 'assigned' to these populations computationally (Fig. 4d). Interestingly, Em-Ex-derived X-cells were highly enriched for cells assigned to FL HSC. Indeed, 70 and 6% of Em-Ex-derived X-cells were assigned to E12.5 FL HSC and E14.5 FL HSC (CD45+CD150+CD48−CD201+), respectively (Fig. 4d). These data suggest that X-cells are enriched for developmentally more mature hematopoietic progenitors[43].

Interestingly, the vast majority of profiled YS-Ex cells (92%) were assigned to E11.0-T2-pre-HSC, suggesting developmental immaturity relative to Em-Ex cells (Fig. 4d). A comparison of E11.0-T2-pre-HSC and E12.5-FL HSC revealed 353 down-regulated genes and 80 upregulated genes (FDR<0.05 and $Log_2$ fold change <−2 or >2, respectively; eWrs-test and Benjamini and Hochberg $p$-value adjustment) (Fig. 4e). Remarkably, compared to other analyzed cells in our scRNAseq dataset, X-cells showed lower expression for the downregulated genes and higher

**Fig. 4** scRNAseq analysis identifies a rare HSC-like population specific to embryo explant cultures. **a** Limiting dilution transplantation of Em-Ex cells. One, two or three embryo equivalents were transplanted/recipient. Data pooled from three independent experiments. Left panel: Experimental schematic. Middle panel: %CD45.2$^+$ PB of recipients at 16 weeks post-transplant. Each circle represents an individual recipient. Red circles highlight non-engrafted recipients. The number of mice engrafted/number of recipients at each cell dose is shown. Right panel: $\chi^2$ analysis revealed a fit to the limiting dilution model (p-value = 0.717; t-test). The Log of the non-responding fraction is shown for each cell dose. **b** Representative FACS plots showing sorting strategy for CD31$^+$CD45$^+$c-Kit$^+$CD41$^-$ cells in YS-Ex and Em-Ex after 10 days of culture. 49 E8.5 concepti (5–7sp) were dissected, cultured and cells of interest collected by FACS. Post-sort purity check of sorted populations are shown. **c** Single cell global gene expression analysis of Em-Ex- and YS-Ex-CD31$^+$CD45$^+$c-Kit$^+$CD41$^-$ cells. Projection of single cell gene expression profiles onto tSNE1 (X-axis) versus tSNE2 (Y-axis) with Em-Ex cells in pink and YS-Ex cells in blue. X denotes the Em-Ex sub-population circled in red. (n = 3504 Em-Ex cells; n = 3037 YS-Ex cells) **d** Maturation stage analysis. Identified population was compared to available profile expression datasets on embryonic pre-HSC and HSC from Zhou et al.[42]. Assignment of cultured YS-Ex, Em-Ex and X-cells to embryonic hematopoietic populations. From the AGM: E11_EC (E11-Endothelial cells; CD31$^+$VE-cadherin$^+$CD41$^-$CD43$^-$CD45$^-$Ter119$^-$); E11_T1 (E11-pre-HSCs; CD31$^+$CD45$^-$CD41$^{low}$c-Kit$^+$ CD201$^{high}$); E11_T2_CD201neg (E11-pre-HSCs; CD31$^+$CD45$^-$CD41$^{low}$c-Kit$^+$CD201$^-$); E11_ T2 (E11-AGM $^-$CD31$^+$CD45$^+$CD41$^{low}$ CD201$^{high}$); from the fetal liver, E12.5_FL (HSCs, Lin$^-$Sca-1$^+$Mac-1$^{low}$CD201$^+$); E14.5_FL (HSC, CD45$^+$CD150$^+$CD48$^-$CD201$^+$); and Adult HSC (BM-HSC, CD45$^+$CD150$^+$CD48$^-$CD201$^+$). **e** Volcano plot of differentially expressed genes between E11_T2 and E12.5_FL. Downregulated genes are shown in blue and upregulated in red, (FDR<0.05 and Log$_2$ fold change <−2 or >2, respectively. **f** Pattern of the 80 most differentially expressed genes in X-population. Source data are provided as a Source Data file

expression for the upregulated genes (Fig. 4f). Upregulated genes were enriched in proliferation related functional terms (i.e., E2F cell cycle targets and DNA-repair genes (FDR $q$-values = 3.98 × $10^{-3}$ and 1.58 × $10^{-2}$, respectively; Fisher's exact test and Benjamini and Hochberg p-value adjustment). This is consistent with increasing proliferation as HSC transition from the AGM to the FL[1]. Genes defining epithelial-to-mesenchymal down-regulated in the E11.0-pre-HSC to E12.5-HSC transition were also downregulated in the X-cells (FDR $q$-value = 2.85 × $10^{-6}$; Fisher's exact test and Benjamini and Hochberg $p$-value adjustment). Of note, multiple cell surface proteins appear selectively upregulated in Em-Ex X-cells (e.g., *Cxcr4, Mpl, Il6ra*, and *Tnfrsf1*, Supplementary Fig. 2). Further work will be required to determine if these cell surface markers hold functional utility for further enrichment of transplantable dHSC from Em-Ex. Indeed, identifying a specific marker for embryonic populations with *bona fide* dHSC potential could be very useful for in vitro screens.

**Intraembryonic LYVE1$^+$ blood precursors yield dHSC activity.** Our studies suggest that the pre-circulation and peri-circulation YS is not a major source of dHSC. However, it was recently reported that >40% of adult blood may emerge from the early YS and vitelline endothelium[14]. Lee and colleagues used *Lyve1-eGFP-hCre* knock-in mice (*Lyve1$^{Cre}$*) (Fig. 5a) to claim an extra-embryonic origin for a significant fraction of adult blood, but not primitive erythrocytes or the definitive hematopoietic precursors of the embryo, placenta or umbilical vessels. LYVE1 expression was reportedly highest in the endothelium of the YS and vitelline vessels (VV) in the mid-gestation conceptus[14].

We used *Lyve1$^{Cre}$*, *Rosa26$^{+/mTmG}$* and *Rosa26$^{+/Confetti}$* mice to further examine the contribution of LYVE1$^+$ cells to lifelong hematopoiesis. The unrecombined *Rosa26$^{mTmG}$* allele ubiquitously labels all cells with the tdTOMATO protein (Fig. 5a). CRE recombinase (CRE) drives the excision of the *tdTomato* cassette and stably labels CRE$^+$ cells and their progeny with GFP (Fig. 5a)[44]. The unrecombined *Rosa26$^{Confetti}$* multicolored cassette does not drive any fluorescent expression. Here, CRE induces recombination of the *Rosa26$^{Confetti}$* allele, which results in cells and their progeny stably labeled with one of four fluorescent proteins (GFP, CFP, RFP, or YFP, Fig. 5a)[45]. We observed ≈20 and ≈5% labeled cells in the adult blood of *Lyve1$^{+/Cre}$Rosa26$^{+/mTmG}$* and *Lyve1$^{+/Cre}$Rosa26$^{+/Confetti}$* mice, respectively (Fig. 5b), confirming that cells expressing CRE via the *Lyve1$^{Cre}$* allele at some point during their development contribute to adult hematopoiesis, albeit in our hands less than the 40% previously

reported[14]. Similar to previous reports, we observed that the endogenous eGFP signal from the *Lyve1$^{Cre}$* knock-in allele (Fig. 5a) was faint and did not hinder detection of the eGFP signal from the *Rosa26$^{mTmG}$* reporter allele[14,46].

To further interrogate LYVE1 as a marker of early hematopoietic progenitors, we next assessed LYVE1 expression in post-circulation concepti. We observed both LYVE1$^{low}$ and LYVE1$^{high}$ cells in E9.5 (19–22sp) and E10.5 (32–36sp) YS and embryos (Fig. 5c–f). Most embryo LYVE1$^+$ cells were LYVE1$^{low}$ (Fig. 5c–f). Indeed, at E9.5 and E10.5 the absolute numbers of embryo- and YS-LYVE1$^{low}$VE-Cadherin$^+$CD31$^+$CD41$^+$c-Kit$^+$ cells were not significantly different (Fig. 5d, f). Thus, LYVE1 expressing cells are readily detectable in post-circulation embryos and YS. Further, examination of public gene expression databases revealed *Lyve1* mRNA in dHSC precursors in E11-E11.5AGM (Supplementary Fig. 3)[42,43]. Altogether, these data confirm that LYVE1 is expressed by putative hematopoietic progenitors in both YS and embryonic tissues post-circulation[14].

Next, we examined LYVE1 expression in peri-circulation (5–7sp concepti) mouse concepti using flow cytometry and confocal microscopy. We again observed distinct LYVE1$^{high}$ and LYVE1$^{low}$ populations in both YS and embryos (Fig. 6a). Although both populations were more abundant in YS relative to embryos, especially LYVE1$^{high}$ cells, LYVE1 expressing cells were clearly detectable by flow cytometry in 5–7sp embryos (% of YS-LYVE1$^{high}$ = 2.86; % of Em-LYVE1$^{high}$ = 0.11; % of YS-LYVE1$^{low}$ = 3.11; % of Em-LYVE1$^{low}$ = 0.56) (Fig. 6a, b). Most LYVE1$^{high}$ and LYVE1$^{low}$ cells in both 5–7sp YS and embryos also expressed CD41 (≈70 and ≈85%, respectively, Fig. 6a, c). Moreover, LYVE1$^+$CD31$^+$ cells were apparent in the endothelium of the paired aorta of E8.5 embryos (Fig. 6d). Altogether, these data demonstrate that LYVE1 expression is not restricted to YS in peri-circulation embryos.

To gain further insight into the fidelity of LYVE1 as a specific marker of YS and vitelline hemogenic endothelium, YS and embryos were collected at the onset of the heartbeat from 5–7sp-*Lyve1$^{+/Cre}$Rosa26$^{+/mTmG}$* concepti (Fig. 7a). Here, GFP$^+$ cells were readily detected in YS (Fig. 7b, c), with an average frequency of 3.9% (Fig. 7c). Labeled cells were much less frequent in 5–7sp embryos (average labeling of 0.067%, Fig. 7b, c). We previously observed that the vast majority of LYVE1$^+$ cells in *Lyve1$^{+/+}$Rosa26$^{+/+}$* concepti are CD41$^+$ (Fig. 6c). Similarly, most GFP$^+$ cells in 5–7sp *Lyve1$^{+/Cre}$Rosa26$^{+/mTmG}$* concepti were CD41$^+$ (Fig. 7b, d). Additionally, the %CD41$^+$ cells was about 20 times larger in the YS than in the embryo (Fig. 7e). As CD41 labels hematopoietic precursors at this early developmental stage, we focused on the recombination efficiency in the CD41$^+$

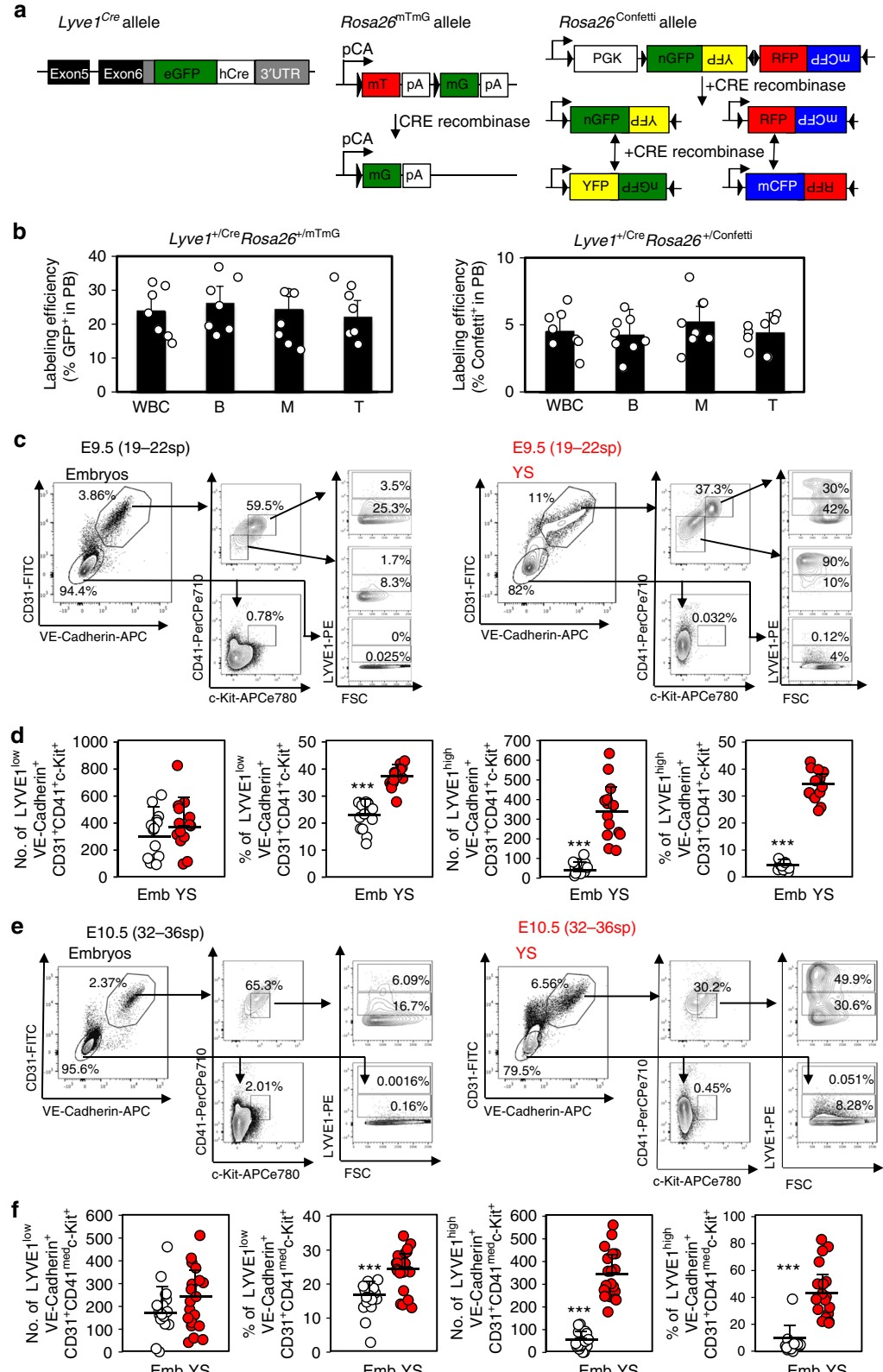

compartment and found it to be about 5% in the YS and 1.1% in the embryo (Fig. 7f)[30,32–34,36,37]. Importantly, GFP+ cells in E8.5 embryos were confined to the endothelium of the paired aortas (Fig. 7g).

As our explant culture system allows us to visualize the emergence of phenotypic hematopoietic precursors from pre-circulation and peri-circulation concepti, we next examined GFP labeling in cultured YS-Ex and Em-Ex-derived hematopoietic precursors (CD31+CD45+c-Kit+CD41−). We observed that, on average, 61% of YS-Ex-derived and 47% of Em-Ex-derived CD31+CD45+c-Kit+CD41− cells were GFP+ (Fig. 7h, i). Lethally irradiated CD45.1+CD45.2+ recipients were next transplanted with cells isolated from Lyve1+/CreRosa26mTmG 5–7sp Em-Ex and 2 × 10^5 CD45.1+ WBM cells (Fig. 7j-l). As expected from our

**Fig. 5** LYVE1 is expressed by E9.5-E10.5 putative hematopoietic progenitors in embryonic and YS tissues. **a** Left panel: Schematic of *Lyve1⁻eGFP-hCre* knock-in allele (*Lyve1^Cre^*). Adapted from ref. [41]. Middle panel: Schematic of the *Rosa26*^mTmG^ allele[39]. The membrane-targeted tandem dimer tomato (mT) protein is expressed from the unrecombined allele under the chicken *β-actin* core promoter with a CMV enhancer (pCA). Cre expression excises the mT cassette and results in expression of the membrane-targeted enhanced green fluorescent protein (mG). mT protein pA indicates polyadenylation sequences and black triangles are loxP sites. Right panel: Schematic of *ROSA26*^Confetti^ allele. Unrecombined allele does not yield the expression of any fluorescent protein. Cre recombinase expression results in a two-steps recombination event that renders the expression of one of the four *Confetti*-colors (GFP, YFP, RFP, or CFP)[40]. **b** Left panel: %recombination (i.e., GFP⁺) in *Lyve1^+/Cre^Rosa26^+/mTmG^* PB at 2 months of age (*n* = 8). Right panel: % recombination (i.e., %Confetti + ) in*Lyve1^+/Cre^Rosa26^+/Confetti^* PB (*n* = 8). Labeling efficiencies of total white blood cells, B-cells, T-cells and myeloid cells are shown. Average is depicted (error bars indicate ± s.d. of mean). **c–d** Cell surface expression of LYVE1 in YS and embryos dissected from freshly isolated E9.5 (19–22sp) concepti (*n* = 14). **c** Representative flow cytometry plots. **d** The absolute number/concepti and frequency as a percent of total live events of LYVE1^low^ and LYVE1^high^ cells within the VE-Caderin⁺CD31⁺CD41⁺c-Kit⁺ compartment is shown. Two independent litters were analyzed. **e–f** LYVE1 expression in freshly isolated E10.5 (32–36sp) concepti (*n* = 20). The absolute number/concepti and frequency as a percent of total live events of LYVE1^low^ and LYVE1^high^ cells within VE-Caderin⁺CD31⁺CD41^low^c-Kit⁺ cells is shown. Three independent litters were analyzed. Each circle represents an independent YS or embryo. Means and standard deviations are shown. (***$p < 0.001$, two sample *t*-test and eWrs-test; differences held even after multiple testing corrections to control a FDR of 0.05). Source data are provided as a Source Data file

previous experiments (Fig. 1b), 5–7sp *Lyve1^+/Cre^Rosa26*^mTmG^ cultured Em-Ex harbored dHSC activity and reconstituted 15–80% of recipient PB (Fig. 7k). Importantly, 3/4 recipients of E8.5 *Lyve1^+/Cre^Rosa26*^mTmG^-derived Em-Ex displayed GFP⁺ engraftment, demonstrating that *Lyve1*^Cre^ labels the progeny of intra-embryonic HE precursors (Fig. 7l). Indeed, in one recipient, 88% of Em-Ex-derived PB was GFP⁺ (Fig. 7l). Differences in the %GFP⁺ recipient PB reflects the small number of Em-Ex-derived RUs transplanted *per* recipient (3.32 RUs = 4EE*0.83RUs per explant, Fig. 4a) and low recombination efficiency of the *Rosa26*^mTmG^ allele (about 20%, Fig. 5b). We also tested cells isolated from 19–22sp *Lyve1^+/Cre^Rosa26*^mTmG^ Em-Ex and YS-Ex for dHSC activity via transplantation. Here, Em-Ex-derived PB reconstitution was high (averaging 51%, *n* = 10). Although the % GFP⁺ PB varied widely (Fig. 7l), 8/10 recipients displayed GFP + PB reconstitution. As expected (Fig. 1c, d), *Lyve1^+/Cre^Rosa26*^mTmG^ YS-Ex did not engraft recipient mice (Fig. 7k, l). In sum, our data reveal that the embryo harbors LYVE1⁺ blood precursors capable of giving rise to dHSC.

## Discussion

Identifying the original site of dHSC specification during embryogenesis has been a major challenge due to the migratory nature of blood precursors in the circulating embryo and the absence of assays to rigorously test hematopoietic potential of pre-circulatory tissues. Here, we report the robust generation of dHSC activity from pre-circulation and peri-circulation murine embryos. dHSC activity could only be detected in cultured embryo explants and not cultured YS explants, indicating that the YS is unable to autonomously generate dHSC activity, at least under the conditions tested here. This implies that either the YS does not contain dHSC precursors or that it lacks the essential support environment for maturation of such precursors. Our findings are consistent with those of Cumano and colleagues, but in those studies, the authors were only able to detect very low levels of HSC activity from pre-circulation intra-embryonic tissues (<5%), and only after transplantation into immunocompromised *Rag2γc⁻/⁻* recipients[12]. Poor reconstitution activity left open the possibility that YS harbored dHSC activity that fell below their detection threshold. Indeed, very recent studies suggest an extraembryonic (i.e., YS and/or vitelline hemogenic endothelium) origin for a large fraction of lifelong hematopoiesis[13,14]. However, the high levels of dHSC activity detected in our culture system (up to >75% reconstitution of PB after transplantation into immunologically competent mice) allows us to rigorously compare YS and embryos and demonstrate that pre-circulation YS lacks significant dHSC potential

(Fig. 1b). None-the-less, our data support a model in which dHSC originate from intra-embryonic precursors in pre-circulation and peri-circulation concepti.

CD31⁺CD45⁺c-Kit⁺CD41⁻cells harbored all Em-Ex-derived dHSC activity in our experimental system. The development of this population required exogenous administration of SCF and IL-3 (Fig. 3). This agrees with previous studies where SCF is also required for the maturation of E9.5 pro-HSCs and E10.5 type I and type II pre-HSCs. *Scf* is expressed in the E9.5 dorsal aorta and upregulated at E10.5 when HSPC specification accelerates[29–31]. *Scf*⁻/⁻ mice display reduced dHSC activity and peri-natal death[39]. IL-3 induces maturation of dHSC between E10.5 and E11.5 in the AGM[40,41]. In contrast, exogenous administration of FLT3L is dispensable for dHSC maturation ex vivo[31]. Our finding that 5–7sp Em-Ex also require SCF and IL-3 to effect dHSC maturation ex vivo suggests that even very early dHSC precursors require these same signals.

Interestingly, CD31⁺CD45⁺c-Kit⁺CD41⁻ cells also emerged in cultures of early YS (Fig. 4b), where they clearly lack detectable dHSC activity (Fig. 1b–d). The ability of our culture conditions to yield bona fide dHSC from pre-circulation concepti provides a unique opportunity to study YS-specific and embryo-specific hematopoiesis and the critical molecular features that allow Em-Ex to render dHSC activity. This could be useful for efforts to derive bona fide dHSC from alternative sources ex vivo and provide specific reporters for in vitro screening. Remarkably, single cell global gene expression profiling identified an Em-Ex-derived population (X-cells, Fig. 4c) that does not emerge from YS cultures and displays enrichment for genes expressed by dHSC found in the FL (Fig. 4d). Further analysis identified multiple cell surface molecules selectively expressed by this unique population (e.g., *Cxcr4, c-Mpl, Il6ra,* and *Tnfrsf1,* Supplementary Fig. 2). These cell surface markers harbor putative utility to both refine the Em-Ex population harboring dHSC activity and explore for in vivo surrogates that may represent critical transitions in the maturation of dHSC in vivo. Exploring the biological significance of these and other markers will be the subject of future investigations.

An alternative approach to defining the embryonic origins of lifelong hematopoiesis is the use of YS versus embryo-specific lineage markers. This approach has proved challenging as YS and embryo-derived hematopoietic precursors exhibit similar developmental programs[1]. Both are mesodermal in origin and emerge from endothelial precursors. Previously, temporally controlled *Runx1⁻*based lineage tracing suggested YS-derived contribution to adult hematopoiesis[13]. However, the duration of the labeling pulse was unclear and only a low percentage of adult blood cells were labeled[1,13]. Additionally, RUNX1⁺ cells were apparent

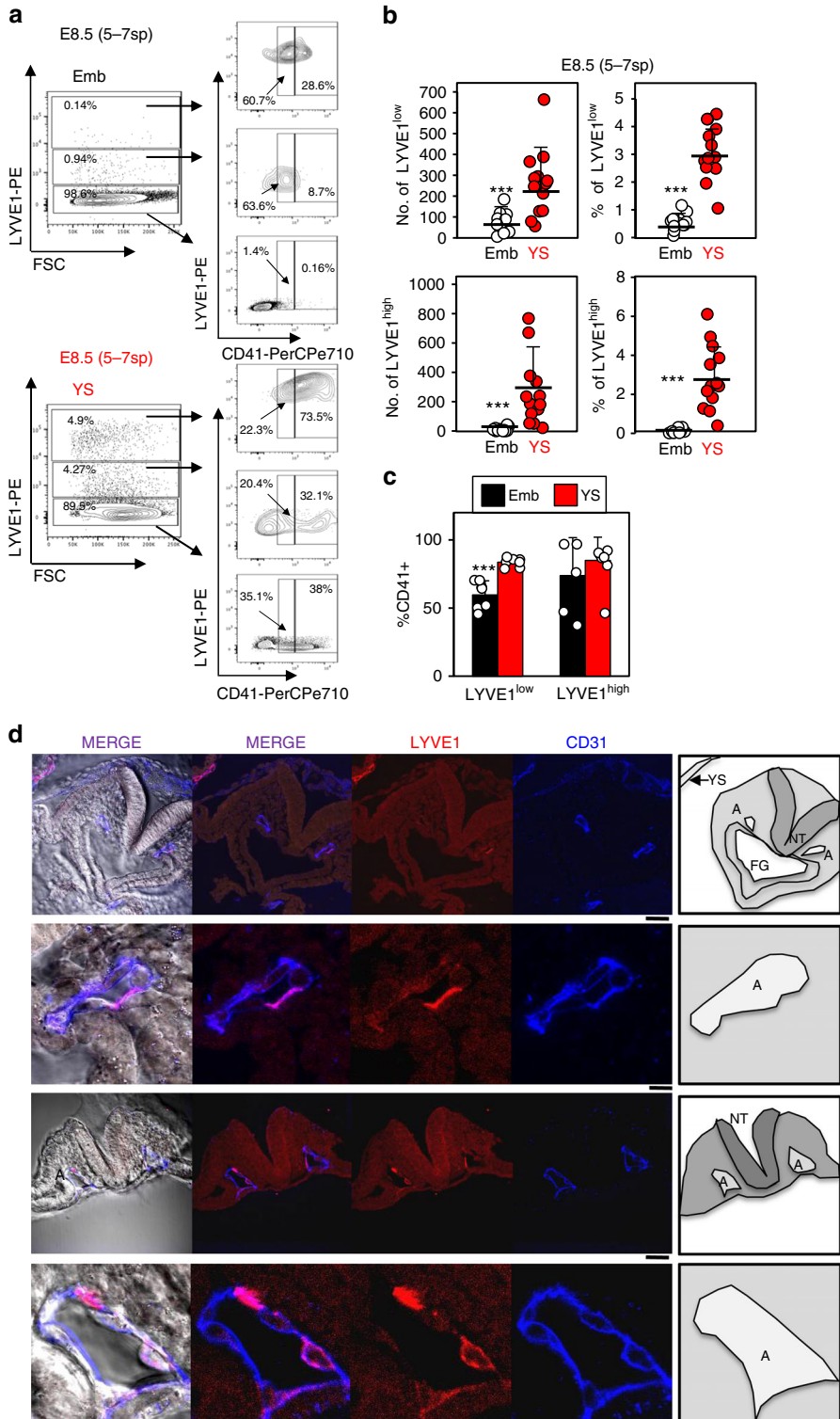

**Fig. 6** LYVE1 is expressed by both yolk sac and embryo cells at E8.5. **a** Representative flow cytometry plots of LYVE1 expression in freshly isolated E8.5 (5–7sp) embryos and YS. **b** Cell surface expression of LYVE1 in YS and embryos dissected from freshly isolated E8.5 (5–7sp) concepti ($n = 16$). The absolute number per concepti and frequency as a percent of total live events of LYVE1[low] and LYVE[high] cells is shown. Each circle represents an independent YS or embryo. **c** Frequency of CD41[+] cells as a percent of total live events of LYVE1[low] and LYVE1[high] cells is shown ($n = 16$) for embryos and YS. **a**–**c** Two independent litters were analyzed. (***$p < 0.001$, two-sample t-test for LYVE1[low] and eWrs-test LYVE1[high]; differences held even after multiple testing corrections to control a FDR of 0.05). Means are shown. Error bars denote standard deviation. **d** Representative confocal images showing the pattern of expression of LYVE1 in E8.5 embryos and YS. Two independent embryos are shown. For each representative embryo an inset zoomed on one of the paired aortas is also shown. Anti-LYVE1 conjugated to CF[TM]568 is shown in red and anti-CD31[−]BV421 is shown in blue. Scale bars stand for 50 µm for the low magnification (i and iii) and 10 µm for the high magnification (ii and iv). Schematics indicate anatomical parts: FG: foregut; A: paired-aorta; NT: Neural tube. Source data are provided as a Source Data file

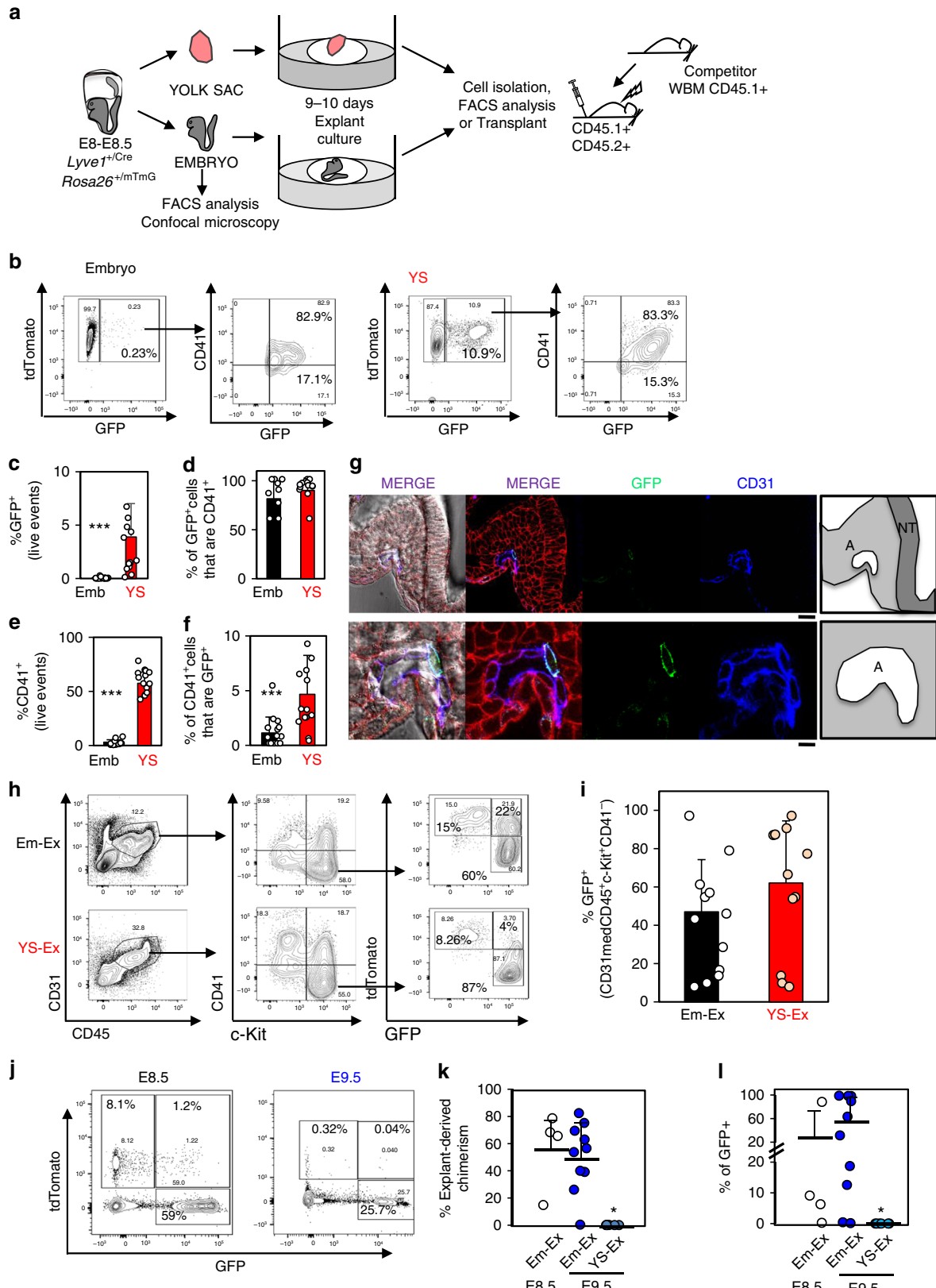

outside the YS, raising doubts that all *Runx1*-induced Cre recombinase activity in early embryos was YS-restricted[1,13]. Further, this study employed a *Cre recombinase* knock-in mouse model that was hemizygous for *Runx1*. *Runx1*[−/+] mice are severely perturbed in both the timing of hematopoietic specification and the cell surface phenotype of emerging hematopoietic

progenitors[47]. Thus, observations made in a hemizygous *Runx1* mouse model are unlikely to reflect normal steady-state hematopoietic ontogeny.

It is possible that the YS harbors dHSC precursors that fail to 'mature' in our system due to the absence of key embryo-derived non-cell-autonomous cues. In this scenario, YS-precursors may

**Fig. 7** Blood precursors emerge from intra-embryonic LYVE1$^+$ cells at the onset of the heartbeat. **a** Experimental schematic. YS and embryos dissected from CD45.2$^+$ E8.5 (5–7sp) *Lyve1*$^{+/Cre}$*Rosa26*$^{+/mTmG}$ concepti were either analyzed by flow cytometry for *Lyve1*-Cre-dependent labeling (GFP$^+$ cells), cultured as explants for 10 days to examine *Lyve1*-Cre-dependent labeling in emerging hematopoietic populations or transplanted. **b–g** Flow cytometry analysis and confocal microscopy of freshly isolated E8.5 (5–7sp) *Lyve1*$^{+/Cre}$*Rosa26*$^{mTmG}$ YS and embryos. Non-recombined *Rosa26*$^{+/mTmG}$ cells are tdTomato$^+$ and recombined *Rosa26*$^{+/mTmG}$ cells are GFP$^+$. **b** Representative flow cytometry plots. **c** Average % recombination (i.e., GFP$^+$) of total YS or embryo cells. ($n = 14$) from three independent litters. **d** % of total GFP$^+$ cells expressing CD41. **e** %CD41$^+$ in total cells. **f** % recombination (i.e., GFP$^+$) in CD41$^+$ cells. **g** Representative confocal microscopy images. GFP (green), tdTomato (red). Endothelial cells (blue, labeled by anti-CD31-BV421). An inset of a paired-aorta is shown. Scale bars: 50 µm and 10 µm. Schematics: A: paired-aorta; NT: Neural tube. **h–i** Flow cytometry analysis of E8.5 (5–7sp) *Lyve1*$^{+/Cre}$*Rosa26*$^{mT/mG}$ YS-Ex and Em-Ex after 10 days of culture. **h** Representative flow cytometry analysis of tdTomato and GFP labeling in YS-Ex and Em-Ex-derived CD31$^+$CD45$^+$c-Kit$^+$CD41$^-$ cells. **i** Quantification of *Lyve1*-Cre dependent GFP labeling in CD31$^+$CD45$^+$c-Kit$^+$CD41$^-$ cells. Each dot represents an independent conceptus. Bars represent average %recombination ($n = 12$) from three independent litters. **j–l** PB of recipients transplanted with cultured E8.5 (5–7sp) or E9.5 (19–22sp) *Lyve1*$^{+/Cre}$*Rosa26*$^{mT/mG}$ Em-Ex or YS-Ex. **j** Representative flow cytometry plots of recipient PB. tdTomato and GFP labeling is shown. **k** % Explant-derived PB, estimated as the sum of the % of tdTomato$^+$ and % of GFP$^+$ cells in the PB. Each circle represents an independent recipient. **l** %*Lyve1*-Cre dependent GFP labeling estimated as %GFP$^+$ amongst % of Explant-derived PB cells is shown. Each circle represents an independent mouse transplanted with four ee of E8.5 CD45.2$^+$ *Lyve1*$^{+/Cre}$*Rosa26*$^{mTmG}$ cultured Em-Ex ($n = 4$) in three independent experiments or one ee of E9.5 CD45.2$^+$ *Lyve1*$^{+/Cre}$*Rosa26*$^{mTmG}$ cultured Em-Ex ($n = 10$) or YS-Ex ($n = 6$) in other three independent experiments. (*$p < 0.05$; ***$p < 0.001$; eWrs-test). Means and standard deviations are shown. Source data are provided as a Source Data file

need to migrate to a maturation niche in the embryo, as proposed by Matsuoka and colleagues[18]. Indeed, *Runx1*$^+$*Gata-1*$^-$ (hemogenic angioblasts) have been reportedly observed migrating from the E7.5 extraembryonic YS into the embryo proper in cultured embryo explants, where they were proposed to receive a currently undefined signal that induces competence to contribute to lifelong hematopoiesis[19]. However, here again the duration of the 4-OHT labeling pulse was unclear, raising the possibility that rare *Runx1* + intraembryonic precursors could have been labeled between E8 and E8.5. Further, it has not been explicitly demonstrated that *Runx1*$^+$*Gata-1*$^-$ cells seen migrating in ex vivo cultures from the extra-embryonic to the intra-embryonic region are capable of contributing to lifelong hematopoiesis in vivo. In our explant system, pre-circulation and peri-circulation YS explants failed to generate any dHSC activity.

Although, *Lyve1*$^-$Cre lineage tracing was recently employed to discern blood derived from YS versus embryonic endothelial precursors[14], we observed two distinct LYVE1$^+$ populations (LYVE1$^{high}$ and LYVE1$^{low}$) in both YS and embryos isolated from E8.5, E9.5 and E10.5 concepti (Figs. 5–6). Although the frequency and absolute numbers of LYVE1$^{high}$ cells is higher in the YS, the absolute numbers of LYVE1$^{low}$ cells between YS-derived and embryo-derived blood precursors is not significantly different (Figs. 5–6). Moreover, we observed LYVE1$^+$ endothelial cells in the endothelium of the paired embryo aortas prior to the onset of circulation (Fig. 6d and Fig. 7g). Furthermore, *Lyve1*$^{+/Cre}$*Rosa26*$^{+/mTmG}$ embryos isolated at the onset of the first heartbeat (5–7sp) and cultured as explants gave rise to GFP$^+$CD31$^+$CD45$^+$c-Kit$^+$CD41$^-$ cells, which is the transplantable population that emerges in this system (Fig. 3c and Fig. 7h, i). On average, about 50% of emerging CD31$^+$CD45$^+$c-Kit$^+$CD41$^-$ cells were GFP$^+$ in these cultures, with one Em-Ex yielding 100% GFP$^+$CD31$^+$CD45$^+$c-Kit$^+$CD41$^-$ cells (Fig. 3c and Fig. 7h, i). Most importantly, transplantation of 5–7sp *Lyve1*$^{+/Cre}$ *Rosa26*$^{+/mTmG}$ Em-Ex revealed that LYVE1$^+$ embryo-derived progeny consistently contributed to the engraftment of lethally irradiated recipients (Fig. 7j–l). Thus, embryo-derived hematopoietic precursors must also express LYVE1. Altogether, these data confirm LYVE1 as an early marker of definitive hematopoiesis, reveal that LYVE1 expression is not restricted to YS hematopoietic precursors and suggest that most LYVE1$^+$ contribution to adult hematopoiesis is embryonic in origin.

In summary, here we report the generation of robust dHSC activity from pre-circulation and peri-circulation embryo explants during ex vivo culture. Cultured YS explants failed to yield dHSC activity under these same conditions, suggesting that the YS contains few precursors capable of maturing into dHSC.

We also present evidence that both YS and embryo-derived hemogenic precursors express LYVE1 during ontogeny. Our study supports a model in which most of lifelong hematopoiesis derives from hematopoietic precursors that develop de novo in the embryo proper. Future studies will be required to better define the pre-circulation and peri-circulation embryonic precursors that ultimately mature into dHSC. These studies would benefit any effort to coax bona fide dHSC from pluripotent stem cells.

## Methods

**Mice**. C57BL/6 J, C57BL/6.SJL-PtprcaPep3b/BoyJ, *ROSA26*$^{+/mTmG}$ (*Gt(ROSA)26Sor*$^{tm4(ACTB-tdTomato,-EGFP)Luo}$/J), *ROSA26*$^{+/Confetti}$ (Gt(ROSA)26Sor$^{tm1(CAG-Brainbow2.1)Cle}$/J) and *Lyve1-eGFP-hCre* knock-in mice (*Lyve1*$^{Cre}$) mice were acquired from The Jackson Laboratory (Bar Harbor, Maine) and housed in a pathogen-free facility. Both females and males were used. All animal experiments were carried out according to procedures approved by the St. Jude Children's Research Hospital Institutional Animal Care and Use Committee and comply with all relevant ethical regulations regarding animal research.

**Genotyping**. Polymerase chain reactions (PCR) were performed using Go Taq DNA Polymerase (Promega, Madison WI) as indicated by the manufacturer. PCR conditions: (95 °C, 2');((94 °C, 30''; 56 °C, 30''; 72 °C, 30'')×35); (72 °C, 10').
Primers: *Cre* F1 (5' CTGTTACGTATAGCCGAAAT 3'), *Cre* R1 (5' CTACACCAGAGACGGAAATC T 3'). CRE Positive PCR band: 203 bp. Detection of the GFP allele was used for *mTmG* and *Confetti* allele genotyping: GFP Fw (5' CAGATGAAGCAGCACGACTTCT 3'), GFP Rv (5' AACTCCAGC AGGACCATGTGAT 3'). GFP PCR band: 400 bp.

**Transplants**. For all transplants, 8–12 weeks old CD45.2$^+$/CD45.1$^+$ C57BL/6 J recipients were treated with 11 Gy of ionizing radiation in split doses of 5.5 Gy prior to transplant. Males and females were used as recipients. In each experiment, sexes of the recipient mice were evenly distributed among experimental groups. All cells were transplanted by tail vein injection. E8.5, E9.5, E10.5, and E11.5 CD45.2$^+$ Em-Ex-derived or YS-derived cells were transplanted along with $2 \times 10^5$ CD45.1$^+$ C57BL/6.SJL WBM cells into recipients. For secondary and tertiary transplantation, $5 \times 10^6$, $2 \times 10^6$, $1 \times 10^6$, or $0.8 \times 10^6$ WBM cells were transplanted from primary or secondary recipients. Engraftment was defined as >0.5% CD45.2$^+$ cells in each lineage (T cells, B cells, and myeloid cells) and >1% CD45.2$^+$ total PB.

**PB Analysis**. PB was collected from the retro-orbital plexus in heparinized capillary tubes (Fisherbrand, Pittsburgh, PA) and lysed in red blood cell lysis buffer (Sigma-Aldrich, St. Louis, MO). Cells were stained with the following antibodies: CD45.1-APC (A20) (Biolegend, San Diego, CA) or CD45.1-FITC (A20) (BD Biosciences, San Diego, CA), B220-PECy7 and CD8-PECy7 (53–6.7) (Tonbo Biosciences, San Diego, CA), CD45.2-V500 (104), B220- PerCPCy5.5 (RA3-6B2), Gr1-PerCPCy5.5 (RB6-8C5), Cd11b-PerCPCy5.5 (M1/70) and CD4-PECy7 (RM4-5) (BD Biosciences, San Diego, CA). All antibodies were used at 1:200 dilution. 4',6-diamidino-2-phenylindole (DAPI) staining was used to gate live events. Analysis was performed on a LSR Fortessa and a BD FACSAria III SORP (Special Order Research Product, which contains the following LASERs: 405 nm, 445 nm, 488 nm, 562 nm, and 640 nm) (both BD Biosciences, San Diego, CA) and the data analyzed with FlowJo version 9.4.11 (Tree Star, Ashland, OR). To discern the four Confetti colors: the filter arrangements for each LASER were: CFP, 445 nm LASER —470/24 band-pass filter (BP); GFP, 488 nm LASER—515/20 BP and 505 long-

pass filter (LP); YFP, 488 nm LASER—545/10 BP and 525 LP; RFP, 562 nm LASER—610/20 BP and 600 LP[48].

**Explants cultures and flow cytometry analysis**. Explant culture conditions were adapted from the protocol developed by Medvinsky and colleagues[29]. Embryo and YS-explants were cultured at the air-liquid interface on 0.65 μm DV Durapore Membrane Filters (Merck Millipore, Cork, Ireland) in Iscove's Modified Dulbecco's Medium-Glutamax (IMDM-Glutamax, Thermo Fisher Scientific, Waltham, MA), 20% FCS (Lot #535905; Cat #FB-02; Omega Scientific, Tarzana, CA), 0.1 mM 2-Mercaptoethanol (Thermo Fisher Scientific, Waltham, MA), and 100 units/ml of Penicillin/Streptomycin (Thermo Fisher Scientific, Waltham, MA) supplemented with recombinant murine SCF, recombinant murine IL-3, and recombinant murine FLT3 Ligand (Peprotech, Rocky Hill, NJ, all at 100 ng/ml) (or with other cytokine combinations indicated along the text) for 9–10 days to allow engraftment from E8.5 tissues or for other time periods as detailed in each figure. Membrane filters were sterilized and rinsed three times in boiling water (five minutes/wash). Up to eight explants were carefully placed on each membrane filter. Filters were then placed on in-house-made ring stands (3.175 mm high; outside diameter: 32.3 mm; inside diameter: 23 mm; Teflon) positioned in the well of a non-tissue-culture-treated 6-well-plate (Corning, NY, USA) containing 2.25 mL of the above described culture media. Media was not refreshed during the culture. Explants were recovered from filters using a sterilized scalpel.

Freshly isolated embryonic tissues or cultured explants were dissociated with collagenase (0.0012 g/ml, Sigma-Aldrich, St. Louis, MO) in Phosphate-Buffered Saline (PBS, Thermo Fisher Scientific, Waltham, MA) supplemented with 10% FCS (Omega Scientific, Tarzana, CA). For flow cytometry cell sorting or analysis, dissociated cells were stained with one or more of these antibodies: CD45.2-V500 (104) (BD Biosciences, San Diego, CA), CD41-PerCP-eFluor710 (eBioMWReg30) (eBioscience, San Diego, CA), CD31-FITC (MEC13.3) (BD Biosciences, San Diego, CA), c-Kit-APC-eFluor780 (2B8) (eBioscience, San Diego, CA), Lyve1-PE (#223322) R&D Byosystems, Minneapolis, MN). All antibodies were used at 1:200 dilution. 4',6-diamidino-2-phenylindole (DAPI) staining was used to gate live events. Analysis was performed on a LSR Fortessa and cell sorting on a BD FACSAria III SORP (Special Order Research Product) (both BD Biosciences, San Diego, CA). Cells were collected employing the BD FACSDiva Software (version 8.0.1) (BD Biosciences, San Diego, CA). Data was analyzed with FlowJo version 9.4.11 (Tree Star, Ashland, OR).

**Statistics and reproducibility**. Summary statistics, including mean and standard deviation, were reported for analyses. Two sample $t$ or exact Wilcoxon rank sum test were used to test for a difference between two groups depending on the normality of the data, which was tested by the Shapiro-Wilk test. False discovery rate (FDR) method developed by Benjamini and Hochberg[49] was used to correct for multiple comparisons at a level of 0.05. Otherwise, $p$-values < 0.05 were considered statistically significant. Analyses were conducted in R-3.3.1.

For limiting dilution analysis, parameters were estimated using a generalized linear model with a complementary log-log link. The generalized Pearson Chi-square was used to assess the goodness-of-fit to the model. The analyses were done using program L-Calc and ELDA online software[50].

All in vitro experiments were reliably reproduced at least 3 times with the exception of Fig. 3Eii (the 5 days culture time point was $n = 2$) and Fig. 3Ei (within the 5 days culture time point the SCF/IL3 and the SCF conditions were $n = 2$). Sample size and number of experiment replicates are detailed in each Figure Legend.

**Single cell RNA-seq**. Sorted YS-Ex-CD31$^+$CD45$^+$c-Kit$^+$CD41$^-$ and Em-Ex-CD31$^+$CD45$^+$c-Kit$^+$CD41$^-$ were resuspended in PBS containing 0.04%BSA at a concentration of 500–600 cells/ul. In total, 8600 cells per sample were loaded into each well of a 10X Chromium single cell capture chip with a recovery of 3000–3500 cells per sample. The captured cells then underwent lysis, reverse transcription, cDNA amplification, and library preparation with indexing per the manufacturer's protocol (10X Genomics). The libraries were sequenced together on an Illumina Nextseq500 (Illumina Inc., San Diego, CA) using a high output 150 cycle kit with read lengths recommended by 10X Genomics.

**Single-cell RNAseq analysis**. Raw data processing: Cellranger 2.0.1[51] was used to process the raw fastq files and to generate unique molecular identifiers (UMI) counts for each gene in each cell. The UMI count table was imported into R and analyzed with Seurat 2.1.0 package[52]. The raw counts were normalized using NormalizeData function with default parameters (normalization.method = Log-Normalize). Cells expressing less than 1000 genes, more than 5000 genes or having over 5% mitochondria reads were removed from the analysis to exclude potential dead, broken or doublet cells. Bach effect is removed using R package scran v1.8.1[53] with $k = 20$, sigma = 0.1. To identify the X-cell population, R package Rtsne v0.13[54] was used to reduce data dimension to two for visualization with perplexity = 20, initial_dims = 100. X-cell population was defined by clusters generated with

R package pdfCluster v1.0-2[55] and selected for further analysis. R package scmap v1.1.5[56] was used to map single cells to cell clusters from Zhou et al.[42] with threshold = 0.7. In detail, R package scmap v1.1.5 computed the similarity between expression profile of a single cell and centroid of cell clusters in the reference data[56]. Cells were assigned to reference cell clusters with the highest similarity. Multiple types of correlations (including Cosine similarity and Pearson and Spearman correlations) were calculated and at least one of them had to be larger than 0.7 (i.e., threshold = 0.7).

**CFU assays**. For analysis of CFU potential, 5–7sp embryos and YS were cultured as explants at the air-liquid interface for five or 10 days in the same conditions as indicated above (20% FCS, SCF, IL-3, FLT3L in IMDM). CD31$^+$CD45$^+$ and CD31$^-$CD45$^-$ cells were isolated by FACS and plated in M3434 methylcellulose (STEMCELL Technologies, Vancouver, Canada). Approximately 60,000 sorted cells were dispensed to 3 mL of M3434 and equally distributed into two plates (#27150, STEMCELL Technologies, Vancouver, Canada). Colonies were analyzed 10 days after plating.

**Confocal microscopy**. E8.5 concepti were isolated and fixed in 4% paraformaldehyde (Electron Microscopy Sciences, Hatfield, PA) overnight at 4 °C. Concepti were then embedded in 4.5% low-melting-point agarose (Thermo Fisher Scientific, Waltham, MA) and sectioned (100 μm slices) using a vibratome (microtome with vibrating blade VT1200, Leica Byosystems Wetzlar, Germany). Sections were washed three times with PBS 0.3% Triton X-100 (Biorad, Hercules, CA) (30 min/wash) and permeabilized at 4 °C overnight in PBS 0.3% Triton X-100 (Biorad, Hercules, CA). Sections were then blocked in 0.3% Triton X-100 with 3% donkey serum (Sigma-Aldrich, St. Louis, MO) in PBS at 4 °C overnight. Sections were next incubated with previously conjugated anti-LYVE1 (Clone ALY7; eBioscience. San Diego, CA), anti-CD41 (Clone MWReg30; BD Bioscience, San Diego, CA), BV421$^-$Rat Anti-Mouse CD31 (Clone MEC 13.3 (RUO)); BD Biosciences, San Diego, CA) at 1:200 in 1% skimmed milk in 0.3% Triton X-100-PBS overnight. Anti-LYVE1 and anti-CD41 antibodies were conjugated following manufacture's protocol to CFTM568 and CFTM488 fluorophores, respectively, using the Mix-n-StainTM labeling kits (Sigma-Aldrich, St. Louis, MO). Sections were then washed three times with 0.3% Triton X-100 in PBS (30 min/wash). Sections were mounted on Fisherbrand Superfrost Microscope Slides (Fisher Scientific, Pittsburgh, PA) and covered with 22 × 22 mm No.1.0 microscope cover glass (Infolab, Indianapolis, IN). Sections were imaged on a Zeiss LSM780 Confocal Microscope (Zeiss, Oberkochen, Germany) using a ×40, 1.1NA water immersion lens. Images were processed employing a ZEN 2012 software (Zeiss, Oberkochen, Germany).

## Data availability
The authors declare that all data supporting the findings of this study are available within the article and its supplementary information files or from the corresponding author upon reasonable request. Single-cell RNAseq data have been deposited in NCBI's Gene Expression Omnibus (GEO) database under accession code GSE110909. The source data underlying Figs. 1, 2, 3, 4A, 5, 6A–C and 7 and Supplementary Figs. 1A, and 3 are provided as a Source Data file.

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

## Acknowledgements

We thank W. Clements, H. Mikkola, the McKinney-Freeman laboratory and Department of Hematology at St. Jude Children's Research Hospital (St. Jude) for critical discussions and reading of the manuscript; D. Ashmun, S. Schwemberger, and J. Laxton for FACS support; C. Davis-Goodrum, K. Millican, A. Reap, and C. Savage for help with timed pregnancies and transplants. V. Frohlich, S. King, J. Peters and A. Pitre for help with confocal imaging. This work was supported by the American Society of Hematology (S.M.-F.), the Hartwell Foundation (S.M.-F.), the NIDDK (K01DK080846 and R01DK104028, S.M.-F.), the American Lebanese Syrian Associated Charities (ALSAC) (S.M.-F. and St. Jude Cell and Tissue Imaging Center), and the NCI (P30 CA021765-35, SJCRH Cell and Tissue Imaging Center) and the Burroughs Wellcome Fund (C.G.).

## Author contributions

M.G. designed the study, performed and analyzed transplants, collected and analyzed data, and wrote the paper. A.C. analyzed transplanted mice and contributed to study design. X.T. and Y.C. performed analysis on single-cell RNA sequencing data. X.T. wrote relevant sections of the manuscript. S.N., R.C., and C.G. collected single-cell RNA sequencing data. G.K. and W.B. performed statistical analyses, S.M.-F. designed the

study, analyzed data, and wrote the paper. All authors discussed the results and commented on the manuscript.

## Additional information

**Competing interests:** The authors declare no competing interests.

