## [Peer Review File · Nature Communications]

Reviewers' Comments:

Reviewer #1:

Remarks to the Author:

In their manuscript, Ganuza et al present a revised assay with which to define HSC activity from pre- and peri-circulation embryos. Ex vivo culture with critical cytokines, shown by others to support HSC emergence ex vivo, resulted in the emergence of transplantable HSC activity from the early embryo proper, but NOT the yolk sac. The authors determine the putative phenotype of definitive HSCs within the embryo proper and use single-cell gene expression profiling of these phenotypic cells in the embryo and YS to demonstrate that only the embryo has a distinct population of cells with a unique "HSC-specific" gene signature. Lastly, the authors attempt to reconcile their findings with a recent report demonstrating definitive hematopoiesis from the YS using Lyve1-Cre lineage tracing. Using the same model, the authors demonstrate that Lyve1-Cre also marks definitive HSCs in the embryo proper. Together, these data are fairly convincing in demonstrating 1) that the early YS does not contain precursors to definitive HSCs and 2) that Lyve1-Cre labels definitive HSCs in the embryo proper.

The ex vivo culture experiments were performed with numerous controls and experimental conditions (demonstration that culture system maintains transplantable HSCs from later stage YS and embryo explants, manipulation of culture time, etc.) with the aim of demonstrating conclusively that the culture system itself does not preclude HSC activity from YS, but rather that the YS harbors no detectable HSC activity until post-circulation. The analysis of Lyve1-Cre labeling in cultured Em-Ex in Figure 5 is critical and very interesting. Why does the % of Lyve1-Cre labeling increase in Em-Ex HSCs derived from E9.5 to E10.5? Would this be expected? It seems that there is a correspondence between the emergence of Lyve1+ labeling and HSC activity upon transplantation, and the observation of transplants yielding very high chimerism that are also 100% GFP- labeled implies that, within the ex vivo system, MOST HSC activity is from Lyve1-Cre labeled cells. This is a different argument than the one made earlier in the text when assaying the same Lyve1-Cre lineage tracing system in vivo (SFig 5). This discrepancy suggests that, despite the ability to support HSC emergence, the ex vivo culture system may drive HSC maturation or generation via developmental pathways that are not the same as in vivo pathways. This discrepancy should be discussed, since it is on this very basis -- that the development of HSCs in the culture system models endogenous HSC maturation -- that YS HSC activity is excluded.

Other minor points:

p7, line 115 - Engraftment is defined in the methods section as being multilineage (>0.5% in each lineage, M, B, and T). The numbers of engrafted mice described at the bottom on page 7 and the engrafted recipients included in SFig 1Ai don't fit: 5 2-3sp (reconstituted or not?) and only 7 for the 5-7sp group. Please clarify.

At what stage is analysis performed in Figure 2B? Text describes analysis in "early" cultures but figure legend states that concepti were cultured as in Figure 1B. Please clarify.

The statistical analysis being performed in Figures 2E and F is unclear and not described. Are t-tests being used? Most statistical tests do not accurately handle distributions with that many zeros (for Fig 2F in particular), and the differences are generally not that convincing. It may be preferable to discuss these data without statistics.

The analysis in Figure 3 are very interesting but the percent variance explained by each PC (and particularly by PC 9) should be included. If all PCs explain roughly similar variance, then examining PC9 is justified.

At what age are mice being examined SFig4B? Were bone marrow HSC or other populations examined?

Reviewer #2:

Remarks to the Author:

The paper of Ganuza et al., tackles the controversy about the origin of HSC in the developing mouse embryo. Despite wealth of evidence obtained by various groups in various vertebrate models strongly suggesting the intra-embryonic origin of HSC, authors of a few previously published reports argued that HSC arise in the yolk sac.

In their manuscript Dr MacKinney-Freeman with colleagues tackled this issue by two approaches. First, they used a culture to determine whether the embryo body or the yolk sac can generate HSC in vitro. Previously this approach was employed for analysis of E9.5-E11.5 embryos and showed preHSC localisation in the AGM and not in the yolk sac. However, without using pre-circulatory (E8) embryos it was difficult to exclude that preHSC arrived in the AGM from the yolk sac. Ganuza et al., addressed this issue by establishing culture conditions in which E8 pre-circulatory embryos can produce definitive high-level repopulating HSC. They found that only the embryo body and not the yolk sac gives rise to high-level repopulating HSC. This is novel important evidence in support of the intra-body origin of the adult hematopoietic system.

The authors went further and explored in detail another published claim (Lee et al., Cell Reports, 2016), based on speculation that LYVE1 is specific for the yolk sac endothelium that tagging adult blood in LYVE1-Cre mice proves about yolk sac origin of HSC. Ganuza et al., performed a simple check of this model and found that LYVE1 actually labels not only the yolk sac endothelium but also cells in AGM. Furthermore, LYVE1 message was expressed in the population of preHSC in the AGM. It means that LYVE1 Cre –mediated recombination can occur not only in the endothelium of the yolk sac but also in preHSC regardless of their origin. Therefore LYVE1 Cre mediated labelling cannot be used as an indicator of extra-embryonic (yolk sac) site of origin of adult HSC.

Both lines of evidence provided by Ganuza and al. nicely complement each other. These will be very important for the field to clarify the long standing controversy.

My main criticism is that the paper is poorly written. It is unfocused: key messages are often buried among extended descriptions of unimportant details and reasoning. The authors should try to emphasize implications and importance of their two lines of evidence and how they complement each other.

Major comment:

- 1) Exact culture protocol should be described in Methods section. Was the medium ever changed? Was there any difference in repopulation when cultures were kept for 9 and or 10 days?
- 2) This study is interesting but diluted by plenty of insignificant and not important data – e.g. comparison of cytokines and single cell analysis of HSC which emerge after culture. This has little relevance to the major message of the paper. Identification of E8 embryonic precursors of HSC before culture would be much more important.

Minor comments:

- 1) Abstract can be improved by better focussing on the problem rather than on details, e.g. Line 32 – better transition to the sentence “LYVE1 is reportedly...” is needed.
- 2) Although it is not essential for conclusions of the paper, it would be interesting to know whether LYVE1 is expressed in VE-cad+CD41-cells in the yolk sac and in the body of the embryo. I could not find this information.
- 3) Line 99 – perhaps instead of “only observed after transplantation into immunocompromised recipients” should be “after transplantation into immunocompromised Rag2gc^{-/-} recipients” as irradiated wild type mice usually used for analysis of true HSC are also immunocompromised. So this cannot be a critical point as implied by the authors. Rag2gc^{-/-} mice are not standard for a long-term repopulation assay and perhaps can support some low-repopulating cells, which are not real HSC. Clear indication of their genotype where appropriate is necessary.
- 4) It is not clear from serial transplantations, whether individual recipients indicated in the legend means that each of tertiary recipients was reconstituted by cells of a separate primary recipient (through a separate secondary recipient). It needs to be made clear how many primary recipients were used in these experiments and whether their bone marrow cells were pooled. In current form

presenting this massive dataset makes little sense. (This cannot change conclusions of the paper since results of primary long-term transplantations are the most important). Extensive material on secondary and tertiary transplantations - the way they did it - makes the picture very messy and detracts from the main message.

5) Line 150 – the authors discuss the possibility that their culture conditions may not support “YS-derived populations at any time” They say “To test this, YS isolated from later developmental time-points... were cultured...for 5 days prior to transplantation...”.

They were able to repopulate recipients with cultured E11 yolk sacs. However, this does not prove that culture conditions were good for E8 yolk sac. It is also unclear whether yolk sac explants were cleanly separated from extra-embryonic vessels. I suggest that this can be discussed in Discussion section, being in Results it only dilute the main message. Conclusions of the paper will not be affected.

6) Line 171 – in my view this title is confusing and not significant – I would suggest something like: “E8 embryo explants generate CD31+..... dHSC in presence of SCF and IL3”. Their focus on the phenotype of HSC does not bring much novelty as this phenotype is known from previous works and obvious. SCF and IL3 are also well known stimulators of HSC development in the embryo.

7) dHSC – have to be decoded when it appears first in the text.

8) Line 199 – please check statement here- it seems that all three factors gave 18% whereas 2 of them gave 21%.

9) Line 210, Legend for Fig 2 – repopulation after culturing with 3 rather than 1 factor were “marginally statistically higher” – is it really worth discussing this minor observation which is not really very interesting especially because P-value is <0.1)? It is not significant neither by statistical criteria nor biologically.

10) Line 215 – the message of sub-section title is too complicated and a bit confusing. Perhaps instead: “Single-cell transcription profiling reveals HSC-like cell subset generated by Em-Ex but not YS explants”

11) Line 221 – should be made clear at this point that the cells were cultured.

12) How many different samples were used for single cell analysis?

13) For their PCA analysis – what happened when they analysed PC1 vs PC2? Did they see any differences between populations before they came to analysis of PC1 vs PC9?

14) Figure 3B shows PCA analysis – variances (%) should be specified to properly assess the difference between cell subsets.

15) Line 226 – please explain how top genes were selected.

16) It is important to make clearer that the authors analysed phenotype and transcriptome of the output cell populations emerging after culture and not precursors in the embryo which mature into HSC. The authors' writing throughout the text may create an impression of the opposite, since E8 preHSC (analysis of which does not exist in this paper) would be of prime interest for everyone. However identification of E8 preHSC is a difficult task due to low efficiency of this culture system (authors transplanted >4 embryo equivalents per recipient and yet some mice were not reconstituted). This part of the study trying to identify dots on the plot which may be equivalent to HSC is of little interest, because they were identified already functionally. Why is it interesting to identify a number of genes in these cells, which is similar to those published in HSC datasets? It is also not clear why these are compared with bone marrow rather than with foetal liver HSC (also published previously).

17) Line 264 – this statement is correct but strangely formulated. Perhaps can be replaced by: “Putative restriction of LYVE1 expression to YS-endothelium suggests the origin of adult blood from YS-derived LYVE1+ precursors”.

18) Line 270 - %% of LYVE1+ cells should be indicated in the text (not just in the figure legend). Although their % in the body is lower than in the yolk sac – it does not change conclusions of the study but would make it clearer and would set an important focus. Without that presenting %% of CD41+ cells in this paragraph looks as if you are trying to distort the data (which of course is not the case).

19) Line 324 – LYVE1 (not just LYVE)

20) Line 322 – I do not think that these data “strongly suggest”. This is not needed – you can

transition to your functional transplantation experiments without that.

21) I would like to recommend the authors to reduce the amount of text on characterisation of LYVE1 antibody staining – it is too extended and a bit amorphous. This would create a better balance: focus on most important findings. In the end of the day, the fact that you see LYVE1+ cells in the body of the embryo from early stages is an essential and decisive evidence in favour of your conclusions. Presenting too many details and reasoning just dilutes it.

22) Line 337 – transplantation of LYVE1 Cre embryo cells – was it just one experiment? If so, this may require at least one more experiment.

23) Figure 5 shows repopulation with LYV1-Cre x GFP/Tomato cells. The plot shows the presence of GFP+Tom+ double positive cells in the peripheral blood, whereas these are supposed to be mutually exclusive. How do the authors explain this?

24) Line 352 – see comment 3)

25) Line 355 – this last sentence in my view is too strong. Strictly speaking the culture method, even giving robust maturation of dHSC from the body and not from the yolk sac explants, leaves the possibility that this culture is not suitable for the yolk sac. Recently Medvinsky's lab used equal stroma (OP9) for co-culture with both isolated AGM preHSC and equivalent "preHSC-like" YS cells, which actually put these two populations in similar culture conditions and showed that only the AGM precursors could mature into dHSC. If you touch the issue of culture inadequacy for the yolk sac, this should be discussed. Your data are novel but cannot really finally prove the body origin of HSC. You need to discuss differently: your data provide very important supportive evidence for intra-embryonic origin of HSC using peri-circulatory embryos.

26) Related to previous issue: it is very nice that you got robust repopulation but your data show that you produce very few HSC from E8 embryos. The system is robust in that it can generate a few high level repopulating HSC but questionable for its capacity to generate many of them. I see no problem with that but this makes your reasoning (as discussed in 15) even more problematic. Because of that I also think that putting "robust" in the title of the paper is incorrect.

27) I think Discussion can be shorter and better structured if you could consolidate your arguments in a more focused and concise text. At the moment it is too diluted.

28) Line 568 – plated in methylcellulose in which concentration?

29) Figure 1B shows results of 16 independent experiments according to Legend. Approximately 20 mice were transplanted in total. It is a bit unclear. How many recipients were transplanted in each experiment? Did they transplant yolk sac and embryo bodies from the same embryos in same experiments?

30) Figure 2A also shows results of repopulation in 5 independent experiments. Three different populations were transplanted into 5 recipients each. Does it mean that every time you transplanted each population just in one recipient? This requires clarification.

31) Fig 5 Legend – "...estimated as %GFP+ amongst the %of tdTomato+ and % of GFP+ cells" - clarification is required.

Reviewer #3:

Remarks to the Author:

The origin of definitive hematopoietic stem cells (dHSCs) has been debated in the past. Currently, most believe that definitive HSCs are from AGM regions. This is largely based on several previous ex vivo culture studies that demonstrate definitive HSCs can originate from embryo proper (EM) but not yolk sac (YS). Given that the contribution to the definitive hematopoiesis in those assays were low (1-5%), some in the field still think YS may contribute at levels that are very hard to detect. Using lineage-tracing, it has been reported that Lyve+ cells contribute significantly to dHSCs. But the specificity of Lyve expression may not be carefully examined. In this manuscript, Ganuza et al addressed these specific questions by employing several new techniques, including longer-term ex vivo tissue culture and careful examination of Lyve expression. The authors found that EM but not YS are sites where definitive HSCs originate from in ex vivo culture. Using longer-term ex vivo culture, the authors also demonstrated that Lyve+ cells from EM ex vivo culture could generate dHSCs. Overall, the experiments are carefully done and the data are of good quality and

convincing. At present, the manuscript emphasizes too much on the technical aspects of a particular paper (ref 14). I agree with the authors' points. Since this field is controversial in the past and is trending to be settled (EM is the origin of dHSCs), I would suggest the authors emphasize more on novel insights (eg single cell expression analysis) and potential experiments in the future to definitively address the question in vivo. This will help broaden the interest of the current study. Below are specific comments:

1. The longer-culture system allowed the authors to achieve more robust reconstitution from ex vivo cultures. While I appreciate the amount of work and thoroughness, these experiments are not fundamentally different from the earlier experiments. The conclusion is the same from earlier experiments as well. These experiments measure potential rather than what is really happening in vivo. Some discussion on this will be helpful for the field to move forward and provoke more thoughts.

2. It is very interesting that the authors found potential gene signatures for dHSCs in the embryo (Fig. 3). Can any surface antigen from PC9 be used to further purify cells from the culture? What can be done with these genes in the future? Some discussion will really provoke positive thinking and move the field forward.

3. 1C secondary transplant assay, why not transplanting 3-5sp? I am not asking the authors to perform additional transplantation experiments. But some discussion/clarification will be helpful.

Minor points:

1. Line 97, typo '35sp'.
2. Line 295, typo 'Figure 5c'.

Reviewer #4:

Remarks to the Author:

In this paper the authors use transplantation assays in the murine system to address a central long-standing and largely, but not completely, resolved question in the field of developmental hematopoiesis- the intra- vs. extraembryonic origin of hematopoietic stem cells. As the authors point out, the current consensus is that hematopoietic stem cells arise from large arterial vessels of the embryo proper and not from the yolk sac where the first hematopoietic progenitors and maturing blood cells are first found during embryogenesis. However, a recent report using Lyve1 expression (Lee, et al.) comes to the conclusion that 40% of circulating blood cells in the adult arise from an extraembryonic source. The current submission tackles this Lyve1 publication head on and provides evidence for the current consensus that hematopoietic stem cells arise from intra-embryonic sources. As such, current paradigms are not shifted but this paper provides further evidence for an intra-embryonic source of long-term hematopoiesis and represents a significant contribution to the field.

The experiments described in this paper are technically challenging, since they require careful dissection of staged mouse embryos, prolonged explant cultures, and hematopoietic stem cell transplantation studies. This explant culture system was developed in the 1980's and has been used more recently by the Medvinsky laboratory to define pre-hematopoietic stem cell populations in the developing mouse embryo. Importantly, the authors use this explant system to examine mouse embryos just prior to or at the onset of cardiac function and blood cell circulation, that is, prior to significant the intermixing of hematopoietic progenitors and maturing blood cells from the yolk sac into the embryo proper.

The authors show convincingly that long-term hematopoietic stem cell activity emerges from explanted embryo proper and not from explanted yolk sac tissues. They examine the ability of yolk

sac and embryo proper tissues explanted for various periods of time to provide stem cell activity. This provides increased confidence of the negative result for yolk sac and they also find that prolonged explant of E9.5 embryo proper leads to a loss of hematopoietic stem cell activity, pointing to a window in HSC emergence.

Analyzing the explanted cells, they find that hematopoietic stem cell activity is restricted to the CD31+CD45+Kit+CD41- population, consistent with its emergence from hemogenic endothelium. Why was CD31 rather than VE-cadherin, which has been associated with pre-HSC activity, chosen for this analysis? Single cell analysis identifies a small subset of this CD31+CD45+Kit+ population that differs between embryo proper and yolk sac explants, and the expression of genes associated with hematopoietic stem cells provides indirect evidence that this subset may represent bone fide hematopoietic stem cells. What proportion of the CD31+CD45+Kit+ cell population does this subset constitute? Are there any cell surface markers associated with this "Z" subpopulation?

Given the findings of Lee, et al., the authors go on to examine Lyve1 expression in the developing mouse embryo and find its expression within the pre-hematopoietic stem cells of the embryo proper using the immunophenotypic definitions established by the Medvinsky lab. Importantly, the authors show that Lyve1+ cells derived from embryo proper explants were capable of reconstituting adult recipients. Given the conclusions of the Lee paper, were transportation studies performed with Lyve1+ cells derived from yolk sac explants (even though no HSC activity is present in yolk sac explants)? Was Lyve1 expression found in the "Z" single cell population?

Overall the paper is well-written. I would suggest less emphasis in the text on the Lee, et al. paper and the pairing down of some repetitive sentences. This would tighten the paper and make the findings better stand on their own merit, as opposed to their derivation as a "reaction" to the conclusions primarily of the Lee, et al. paper.

Minor points:

1. Given the current evidence that various proportions of many tissue-resident populations in the adult originate from the yolk sac, the statement on line 434 about the potential of the embryo proper to give rise to "all of life-long hematopoiesis" should be toned down.
2. There are 19 genes listed in Figure 3ci.
3. Will the single cell RNA-Seq data be deposited in a public database?

Response to Reviews (Reviewers are listed below followed by our responses in bold)

Response to Reviewer #1:

In their manuscript, Ganuza et al present a revised assay with which to define HSC activity from pre- and peri-circulation embryos. Ex vivo culture with critical cytokines, shown by others to support HSC emergence ex vivo, resulted in the emergence of transplantable HSC activity from the early embryo proper, but NOT the yolk sac. The authors determine the putative phenotype of definitive HSCs within the embryo proper and use single-cell gene expression profiling of these phenotypic cells in the embryo and YS to demonstrate that only the embryo has a distinct population of cells with a unique "HSC-specific" gene signature. Lastly, the authors attempt to reconcile their findings with a recent report demonstrating definitive hematopoiesis from the YS using Lyve1-Cre lineage tracing. Using the same model, the authors demonstrate that Lyve1-Cre also marks definitive HSCs in the embryo proper. Together, these data are fairly convincing in demonstrating 1) that the early YS does not contain precursors to definitive HSCs and 2) that Lyve1-Cre labels definitive HSCs in the embryo proper.

We are pleased that the Reviewer found our work and conclusions convincing. We also thank them for their feedback, which we have used to strengthen our study. We have done our best to address their concerns, as outlined below.

The ex vivo culture experiments were performed with numerous controls and experimental conditions (demonstration that culture system maintains transplantable HSCs from later stage YS and embryo explants, manipulation of culture time, etc.) with the aim of demonstrating conclusively that the culture system itself does not preclude HSC activity from YS, but rather that the YS harbors no detectable HSC activity until post-circulation. The analysis of Lyve1-Cre labeling in cultured Em-Ex in Figure 5 is critical and very interesting. Why does the % of Lyve1-Cre labeling increase in Em-Ex HSCs derived from E9.5 to E10.5? Would this be expected? It seems that there is a correspondence between the emergence of Lyve1+ labeling and HSC activity upon transplantation, and the observation of transplants yielding very high chimerism that are also 100% GFP- labeled implies that, within the ex vivo system, MOST HSC activity is from Lyve1-Cre labeled cells. This is a different argument than the one made earlier in the text when assaying the same Lyve1-Cre lineage tracing system in vivo (SFig 5). This discrepancy suggests that, despite the ability to support HSC emergence, the ex vivo culture system may drive HSC maturation or generation via developmental pathways that are not the same as in vivo pathways. This discrepancy should be discussed, since it is on this very basis -- that the development of HSCs in the culture system models endogenous HSC maturation -- that YS HSC activity is excluded.

Here, the reviewer is expressing concern regarding the apparent differences in the %recombination (*i.e.* %GFP+ labeling) driven by *Lyve1-Cre* in functional HSC derived from E8.5 and E9.5 embryo explants in Figure 5D (the Reviewer actually cites E9.5 and E10.5 but we believe they meant to refer to E8.5 and E9.5, as this is the data displayed in Figure 5 and we did not interrogate GFP labeling at E10.5 in these studies. We apologize if we have mis-interpreted the Reviewer's intent).

We thank the Reviewer for raising this concern. We have interpreted differences in functional output and GFP-labeling of E8.5 *Lyve1*-Cre embryo explants as stochastic. This is because 1) the number of actual functional repopulating units (RUs) transplanted per recipient is very low. We have now performed Limiting Dilution Transplantation Assays with E8.5 Embryo-Explants (please see new Figure 3Aii). Here we estimated 0.8 RUs/E8.5 embryo explant. Thus, in Figure 5D, each recipient of E8.5 embryo explants was transplanted with four embryo equivalents for a total of about 3.2 RUs/recipient. 2) The labeling efficiency of *Lyve*-Cre is poor. If the Reviewer consults Figures 4Ai-ii and 5Bi-ii, they will note that although on average 0.67% of E8.5 embryo-derived cells are LYVE1+, only 0.067% (on average, Figure 5Bii) are GFP+ in *Lyve*-Cre*Rosa26*^{+mTmG} E8.5 embryos. This is labeling efficiency of about 10%. This is consistent with the average labeling efficiency observed in the blood of adult *Lyve*-Cre*Rosa26*^{+mTmG} mice (Supplemental Figure 4Bi).

Further, we have also now transplanted additional recipients (for a total N=10) with E9.5 embryo explants (please see updated Figure 5Dii-iii). As the Reviewer notes, our original dataset could have been interpreted as suggesting that most E9.5-embryo explant derived reconstitution was GFP+, which appeared to conflict with the relatively low level of GFP labeling in the blood of adult *Lyve*-Cre*Rosa26*^{+mTmG} mice (in our original submission, only three E9.5 recipients were shown). However, now the Reviewer can appreciate that the GFP+ reconstitution derived from E9.5-embryo explants is highly variable and most likely reflects variable CRE labeling between explants and the relatively low numbers of transplanted RUs, rather than a biological difference in the developmental pathway employed by maturing HSC in our *ex vivo* culture system versus *in vivo*.

We have attempted to clarify this point in the manuscript (lines 344-348).

Other minor points:

p7, line 115 - Engraftment is defined in the methods section as being multilineage (>0.5% in each lineage, M, B, and T). The numbers of engrafted mice described at the bottom on page 7 and the engrafted recipients included in SFig 1Ai don't fit: 5 2-3sp (reconstituted or not?) and only 7 for the 5-7sp group. Please clarify.

We apologize for this oversight. Supplemental Figure 1Ai now displays the lineage distribution of all but two mice shown in Figure 1B. These two mice (from the 5-7sp group) are not shown because they lack CD45.2+ reconstitution. This has also now been explained in the legend of Supplemental Figure 1Ai.

At what stage is analysis performed in Figure 2B? Text describes analysis in "early" cultures but figure legend states that concepti were cultured as in Figure 1B. Please clarify.

We again apologize for this oversight. The experiments in Figure 2B were indeed performed as in Figure 1B as indicated in the legend of Figure 2B (day 9-10 of culture). We have now incorporated these details into the experimental schematic shown in Figure 2A and eliminated the confusing reference to "early" from the text in line 179.

The statistical analysis being performed in Figures 2E and F is unclear and not described. Are t-tests being used? Most statistical tests do not accurately handle distributions with that many zeros (for Fig 2F in particular), and the differences are generally not that convincing. It may be preferable to discuss these data without statistics.

We thank the Reviewer for raising this point. We apologize for a typographical error on indicating the actual statistical differences in Figure 2F. Differences in the transplants from explants cultured in the absence of growth factors were indeed statistically significant and Figure 2F has now been corrected accordingly. In Figures 2E and 2F and as we indicate in the Methods section, we first tested the normality of the data using the Shapiro-Wilk test and then used two-sample t-test or exact Wilcoxon rank sum test to test the statistical difference between two groups. In this case, the distribution of the data was not normal and so the exact Wilcoxon rank sum test was used to test for the statistical differences between two groups.

The analysis in Figure 3 are very interesting but the percent variance explained by each PC (and particularly by PC9) should be included. If all PCs explain roughly similar variance, then examining PC9 is justified.

We thank the Reviewer for their observation. PC9 explained 3.4% of total variance. This was not surprising since “Z” cell population comprised only 0.4% of the total cell population used for PCA analysis. As such, the small percentage of explained variance by PC9 was expected.

As per additional suggestions from Reviewer 2 (please see point 16 on their comments), we have now updated our analysis of our scRNA dataset to include a comparison with published analyses of the single cell transcriptional profiles of embryonic hematopoietic precursors from throughout ontogeny. As such, we performed alternative analyses and corrections of our own dataset and now display tSNE1 Vs tSNE2 as a means to identify disparate populations within our single cell pool (please see Figure 3C).

At what age are mice being examined SFig4B? Were bone marrow HSC or other populations examined?

We apologize for this oversight. *Lyve1^{+Cre}Rosa26^{+mTmG}* mice were examined at two months of age. This is now included in the legend of Supplemental Figure 4B.

Lee *et al* carefully examined the bone marrow of *Lyve1^{+Cre}Rosa26^{+mTmG}* mice in their manuscript (please see Figure 3G in their paper, Lee *et al. Cell Reports* 2016). We did not examine the bone marrow of adult *Lyve1^{+Cre}Rosa26^{+mTmG}* mice in our study.

Response to Reviewer #2:

The paper of Ganuza et al., tackles the controversy about the origin of HSC in the developing mouse embryo. Despite wealth of evidence obtained by various groups in various vertebrate models strongly suggesting the intra-embryonic origin of HSC, authors of a few previously published reports argued that HSC arise in the yolk sac.

In their manuscript Dr McKinney-Freeman with colleagues tackled this issue by two approaches. First, they used a culture to determine whether the embryo body or the yolk sac can generate HSC in vitro. Previously this approach was employed for analysis of E9.5-E11.5 embryos and showed preHSC localisation in the AGM and not in the yolk sac. However, without using pre-circulatory (E8) embryos it was difficult to exclude that preHSC arrived in the AGM from the yolk sac. Ganuza et al., addressed this issue by establishing culture conditions in which E8 pre-circulatory embryos can produce definitive high-level repopulating HSC. They found that only the embryo body and not the yolk sac gives rise to high-level repopulating HSC. This is novel important evidence in support of the intra-body origin of the adult hematopoietic system.

The authors went further and explored in detail another published claim (Lee et al., Cell Reports, 2016), based on speculation that LYVE1 is specific for the yolk sac endothelium that tagging adult blood in LYVE1-Cre mice proves about yolk sac origin of HSC. Ganuza et al., performed a simple check of this model and found that LYVE1 actually labels not only the yolk sac endothelium but also cells in AGM. Furthermore, LYVE1 message was expressed in the population of preHSC in the AGM. It means that LYVE1 Cre –mediated recombination can occur not only in the endothelium of the yolk sac but also in preHSC regardless of their origin. Therefore LYVE1 Cre mediated labelling cannot be used as an indicator of extra-embryonic (yolk sac) site of origin of adult HSC.

Both lines of evidence provided by Ganuza and al. nicely complement each other. These will be very important for the field to clarify the long standing controversy.

My main criticism is that the paper is poorly written. It is unfocused: key messages are often buried among extended descriptions of unimportant details and reasoning. The authors should try to emphasize implications and importance of their two lines of evidence and how they complement each other.

We thank the Reviewer for their feedback and are pleased that they found our study interesting and an important contribution to the field. We also thank them for their comments on the manuscript and have significantly reorganized it as a result (and as described below).

Major comment:

1) Exact culture protocol should be described in Methods section. Was the medium ever changed? Was there any difference in repopulation when cultures were kept for 9 and or 10 days?

We apologize for not providing enough experimental detail in our original submission. We have now updated our Methods. We now explicitly state that media is not refreshed during the 9-10 days of culture (please see lines 517-518). We further indicate details on how collecting the explants from filters following culture. In our experience, explants cultured 9 or 10 days behave similarly, which is why we have pooled these data in the manuscript.

2) This study is interesting but diluted by plenty of insignificant and not important data – e.g. comparison of cytokines and single cell analysis of HSC which emerge after culture. This has little relevance to the major message of the paper. Identification of E8 embryonic precursors of HSC before culture would be much more important.

We appreciate the Reviewer’s perspective. We fully agree that identifying the E8 hematopoietic precursors would be an impactful discovery. Indeed, in collaboration with Dr. Brandon Hadland from Dr. Irv Bernstein’s laboratory (Fred Hutchinson Cancer Research Center, Seattle WA), who are expert in the dissociation and culture of cells isolated from very young embryos, we performed numerous pilot experiments with this Aim in mind over the past two years. Unfortunately, technical constraints precluded the successful execution of these experiments. Specifically, E8 embryo-derived cells were dissociated, sorted for live cells and then co-cultured with either OP9 stroma or AKT-expressing endothelial cells, which have both been shown to support the maturation of embryonic hematopoietic precursors *in vitro* (Hadland *et al.*, *J Clin Invest*, 2015; Ganuza *et al.*, *Exp Hematol*, 2017). In these experiments, repopulating activity after 9-10 days of culture was not preserved. We also attempted dissociating and re-aggregating live cells sorted from E8-embryos prior to 9-10 days of culture. Again, repopulating activity was not preserved in cultured E8 reaggregates. These experiments were performed both in our laboratory by Dr. Ganuza and also independently by Dr. Hadland in the Bernstein laboratory. Our interpretation of these efforts is that we have not yet identified experimental conditions in which E8-derived hematopoietic precursors are able to withstand the stress of dissociation, cell sorting and/or re-aggregation.

Minor comments:

1) Abstract can be improved by better focusing on the problem rather than on details, e.g. Line 32 – better transition to the sentence “LYVE1 is reportedly...” is needed.

We thank the Reviewer for their feedback. The abstract has now been rewritten to focus on the major question of the study and its findings.

2) Although it is not essential for conclusions of the paper, it would be interesting to know whether LYVE1 is expressed in VE-cad+CD41-cells in the yolk sac and in the body of the embryo. I could not find this information.

We agree that these data would be interesting. E9.5 YS-VE-cad+CD31+CD41- cells were largely LYVE1+. Most of these cells were LYVE1^{high} with a small fraction of Lyve1^{low}. Supplemental Figure 4Ci now shows representative FACS plots of this population. Additionally, our study also now includes immune-fluorescence via confocal microscopy revealing the presence of LYVE1+ cells in E8.5 embryos (see Figure 4B). These cells also express CD31+ and are localized to the endothelium of the paired dorsal aortas. Moreover, we now also show the presence of GFP+CD31+ cells in the endothelial lining of the paired dorsal aorta of E8.5 *Lyve1^{+/Cre} Rosa26^{+/mTmG}* embryos (please see Figure 5Bvi).

3) Line 99 – perhaps instead of “only observed after transplantation into immunocompromised recipients” should be “after transplantation into immunocompromised Rag2gc^{-/-} recipients” as irradiated wild type mice usually used for analysis of true HSC are also immunocompromised. So this cannot be a critical point as implied by the authors. Rag2gc^{-/-} mice are not standard for a long-term repopulation assay and perhaps can support some low-repopulating cells, which are not real HSC. Clear indication of their genotype where appropriate is necessary.

We have edited this statement, as requested, in the current version of the manuscript (now line 101) and have also made edits to a similar statement in line 364.

4) It is not clear from serial transplantations, whether individual recipients indicated in the legend means that each of tertiary recipients was reconstituted by cells of a separate primary recipient (through a separate secondary recipient). It needs to be made clear how many primary recipients were used in these experiments and whether their bone marrow cells were pooled. In current form presenting this massive dataset makes little sense. (This cannot change conclusions of the paper since results of primary long-term transplantations are the most important). Extensive material on secondary and tertiary transplantations - the way they did it - makes the picture very messy and detracts from the main message.

We thank the Reviewer for their input and apologize if our experimental protocols were unclear. In the spirit of focusing the manuscript, we have now moved all data relating to secondary and tertiary transplants from Figure 1 to Supplementary Figure 1. We feel it is important to preserve these data in the manuscript as they demonstrate the extensive self-renewal potential of E8.5 Em-Ex-derived dHSC, which is a key functional criteria for defining *bona fide* transplantable dHSC.

We have also edited the legend of Supplemental Figure 1 to clearly indicate that for secondary transplants, three primary recipients were transplanted independently into three separate cohorts of secondary recipients. We also indicate that for tertiary transplants, bone marrow from four secondary recipients was transplanted independently into four distinct cohorts of tertiary recipients (please see revised legend of Supplemental Figure 1).

5) Line 150 – the authors discuss the possibility that their culture conditions may not support “YS-derived populations at any time” They say “To test this, YS isolated from later developmental time-points... were cultured...for 5 days prior to transplantation...”. They were able to repopulate recipients with cultured E11 yolk sacs. However, this does not prove that culture conditions were good for E8 yolk sac. It is also unclear whether yolk sac explants were cleanly separated from extra-embryonic vessels. I suggest that this can be discussed in Discussion section, being in Results it only dilute the main message. Conclusions of the paper will not be affected.

We thank the Reviewer for this comment and have now removed this statement from the manuscript. Further, as requested, we now also address this point in our Discussion on lines 361 and 370-372.

6) Line 171 – in my view this title is confusing and not significant – I would suggest something like: “E8 embryo explants generate CD31+..... dHSC in presence of SCF and IL3”. Their focus on the phenotype of HSC does not bring much novelty as this phenotype is known from previous works and obvious. SCF and IL3 are also well known stimulators of HSC development in the embryo.

We thank the Reviewer for their suggestion and have modified the title of this section (current line 170). As no previous study (to our knowledge) has reported *robust* repopulating activity from E8.5 embryos, it has not previously been possible to assess if SCF and IL-3 were sufficient to promote the maturation of dHSC from E8.5 hematopoietic precursors. So, although SCF and IL-3 are known to regulate dHSC development, our study is the first to explicitly demonstrate their ability to promote dHSC maturation from E8.5 embryos.

7) dHSC – have to be decoded when it appears first in the text.

With respect, although we did define this acronym on line 25 of the abstract in our original submission, in the spirit of improving the ‘readability’ of our revised manuscript, we now also define it on both lines 25 and 47.

8) Line 199 – please check statement here- it seems that all three factors gave 18% whereas 2 of them gave 21%.

We thank the Reviewer for pointing out this typographical error. It has now been corrected in line 197.

9) Line 210, Legend for Fig 2 – repopulation after culturing with 3 rather than 1 factor were “marginally statistically higher” – is it really worth discussing this minor observation which is not really very interesting especially because P-value is <0.1)? It is not significant neither by statistical criteria nor biologically.

This statement has now been removed from the manuscript and legend of Figure 2.

10) Line 215 – the message of sub-section title is too complicated and a bit confusing. Perhaps instead: “Single-cell transcription profiling reveals HSC-like cell subset generated by Em-Ex but not YS explants”

We thank the Reviewer for their suggestion. The sub-section title has now been changed to “Single-cell transcription profiling reveals an HSC-like population generated by Embryo Explants but not YS explants”.

11) Line 221 – should be made clear at this point that the cells were cultured.

Noted. We have now added the phrase ‘on day 10 post-culture’ to (what is now) line 224.

12) How many different samples were used for single cell analysis?

We apologize for omitting this information. Two samples were processed for scRNA analysis. One sample represents a pool of 49 Embryo-Explants cultured in parallel and the second sample represents a pool of the corresponding 49 YS-Explants. This information has now been added to the legend of Figure 3.Bi.

13) For their PCA analysis – what happened when they analysed PC1 vs PC2? Did they see any differences between populations before they came to analysis of PC1 vs PC9?

We did observe an Embryo-Explant specific population “X” in a PC1 vs PC2 plot. The Embryo specific cell cluster identified from PC1 vs PC2 plot are highly overlapping (~70% cells overlap) with the “X” cell population identified from tSNE plot that we are now including (please see Figure 3C). In response to point 16 below, we have now updated our analysis of our scRNA dataset to include a comparison with published analyses of the single cell transcriptional profiles of embryonic hematopoietic precursors from throughout ontogeny. In this analysis we also performed batch effect corrections designed specifically for single cell RNA-seq to optimize our analysis (please see revised Methods). As such, we performed alternative analyses and corrections of our own dataset and now display tSNE1 Vs tSNE2 as a means to identify disparate populations within our single cell pool (please see Figure 3C).

14) Figure 3B shows PCA analysis – variances (%) should be specified to properly assess the difference between cell subsets.

Please see our response to point 13 above.

15) Line 226 – please explain how top genes were selected.

Please see our response to point 13 above. Additionally, in the updated tSNE analysis, we used all the expressed genes (~6500 genes) to first do the PCA analysis. The top 100 PCs were selected to feed into the tSNE to do dimension reduction.

16) It is important to make clearer that the authors analysed phenotype and transcriptome of the output cell populations emerging after culture and not precursors in the embryo which mature into HSC. The authors’ writing throughout the text may create an impression of the opposite, since E8 preHSC (analysis of which does not exist in this paper) would be of prime interest for everyone. However identification of E8 preHSC is a difficult task due to low efficiency of this culture system (authors transplanted >4 embryo equivalents per recipient and yet some mice were not reconstituted). This part of the study trying to identify dots on the plot which may be equivalent to HSC is of little interest, because they were identified already functionally. Why is it interesting to identify a number of genes in these cells, which is similar to those published in HSC datasets? It is also not clear why these are compared with bone marrow rather than with foetal liver HSC (also published previously).

We thank the Reviewer for their suggestion and have now added numerous comments throughout the manuscript emphasizing that these analyses were of cultured embryo-

explants rather than freshly isolated E8.5 embryos (e.g. lines 215-221, 224, 388-400). As suggested, we have also now compared our scRNA dataset to the global gene expression profiles of fetal liver HSC and additional embryonic hematopoietic precursors across ontogeny (please see Figure 3C-D). We agree with the Reviewer that this is a better approach.

We embarked on this analysis because we wanted to glean insight as to why YS-explants failed to yield dHSC activity in these experiments, despite the fact that a population of CD31+CD45+c-Kit+CD41- cells, which contain all dHSC activity in Embryo-Explants, emerged in these cultures. We suspected that a sub-population might emerge in cultured Embryo-Explants but not YS-Explants, which has proven to be the case. Although we agree that our *in vitro* system likely does not perfectly recapitulate the *in vivo* process, we still think that it might serve as a useful (and pliable) surrogate from which one might glean insight as to pathways and transitions governing the maturation of nascent dHSC during embryogenesis. Indeed, rarely do *in vitro* systems perfectly recapitulation the living organism. Yet, they are still vessels for novel insights that can later be explored *in vivo*.

Here, we have a system that allows us to explore Embryo-specific versus YS-specific hematopoiesis in very early embryos. As our new analyses presented in Figure 3D makes clear, the dHSC emerging from our E8.5 Embryo-Explants are transcriptionally more similar to E12.5 fetal liver HSC than E14.5 fetal liver or adult bone marrow HSC. Thus, our dataset may be useful for identifying novel candidate cell surface markers of dHSC that have recently emerged from the AGM and migrated to the fetal liver. Such molecules could be useful for *in vitro* screens or as reporters in directed differentiation studies. We now discuss this in lines 217-222; 390-400).

17) Line 264 – this statement is correct but strangely formulated. Perhaps can be replaced by: “Putative restriction of LYVE1 expression to YS-endothelium suggests the origin of adult blood from YS-derived LYVE1+ precursors”.

This introductory sentence has now been deleted.

18) Line 270 - % of LYVE1+ cells should be indicated in the text (not just in the figure legend). Although their % in the body is lower than in the yolk sac – it does not change conclusions of the study but would make it clearer and would set an important focus. Without that presenting % of CD41+ cells in this paragraph looks as if you are trying to distort the data (which of course is not the case).

We thank the Reviewer for their suggestion. The %LYVE1+ cells is now clearly stated in the text in lines 308-310 for both embryos and YS.

We have also now added data addressing the physical location of LYVE1+ cells in E8.5 embryos immune-fluorescence via confocal microscopy (Figure 4B, Figure 5Biv, Lines 312-313). In the spirit of focusing the manuscript, all data related to presence of LYVE1+ cells in E9.5 and E10.5 embryos has been moved to Supplemental Figure 4. We also now show

the presence of GFP+CD31+ cells in the endothelial lining of the paired dorsal aorta of E8.5 *Lyve1^{+/-Cre} Rosa26^{+/-mTmG}* embryos (please see Figure 5Bvi and lines 327-328).

19) Line 324 – LYVE1 (not just LYVE)

Corrected.

20) Line 322 – I do not think that these data “strongly suggest”. This is not needed – you can transition to your functional transplantation experiments without that.

Noted and revised (please see line 334-335).

21) I would like to recommend the authors to reduce the amount of text on characterisation of LYVE1 antibody staining – it is too extended and a bit amorphous. This would create a better balance: focus on most important findings. In the end of the day, the fact that you see LYVE1+ cells in the body of the embryo from early stages is an essential and decisive evidence in favour of your conclusions. Presenting too many details and reasoning just dilutes it.

We thank the reviewer for their feedback and have now substantially revised this section of the manuscript.

22) Line 337 – transplantation of LYVE1 Cre embryo cells – was it just one experiment? If so, this may require at least one more experiment.

In our original submission, the *Lyve1^{+/-Cre}Rosa26^{+/-mTmG}* transplantation data presented in Figure 5D represented the following: For E8.5: three independent experiments in which each recipient mouse was transplanted with 4 pooled embryo equivalents of cultured embryo explants. For E9.5: one experiment in which each recipient was transplanted with 1 embryo equivalent of cultured embryo explants. As requested, we have now repeated the E9.5 transplant experiment twice and these data are now also included in Figure 5D (*i.e.* E9.5 transplant data now reflects the results of three independent experiments). The legend also now clearly indicates the number of independent experiments.

23) Figure 5 shows repopulation with LYV1-Cre x GFP/Tomato cells. The plot shows the presence of GFP+Tom+ double positive cells in the peripheral blood, whereas these are supposed to be mutually exclusive. How do the authors explain this?

We thank the Reviewer for pointing out this technical concern. We attribute the tdTomato+GFP+ cells apparent in Figures 5Bi, Ci and Di to a combination of fluorophore protein stability (*i.e.* some residual tdTomato protein present in the cell after allele recombination has been triggered) and leakiness of GFP expression from non-recombined alleles (see Supplemental Figure 4Aii). These tdTomato+GFP+ cells are also apparent in Figure 2E of Lee *et al. Cell Reports* 2016).

24) Line 352 – see comment 3)

We thank the reviewer and have edited line 364 accordingly.

25) Line 355 – this last sentence in my view is too strong. Strictly speaking the culture method, even giving robust maturation of dHSC from the body and not from the yolk sac explants, leaves the possibility that this culture is not suitable for the yolk sac. Recently Medvinsky’s lab used equal stroma (OP9) for co-culture with both isolated AGM preHSC and equivalent “preHSC-like” YS cells, which actually put these two populations in similar culture conditions and showed that only the AGM precursors could mature into dHSC. If you touch the issue of culture inadequacy for the yolk sac, this should be discussed. Your data are novel but cannot really finally prove the body origin of HSC. You need to discuss differently: your data provide very important supportive evidence for intra-embryonic origin of HSC using peri-circulatory embryos.

We appreciate the Reviewer’s perspective and agree that we cannot exclude the possibility that our *in vitro* conditions may not support E8 YS precursors. We have now edited the first paragraph of our discussion to acknowledge this (please see lines 370-372).

26) Related to previous issue: it is very nice that you got robust repopulation but your data show that you produce very few HSC from E8 embryos. The system is robust in that it can generate a few high level repopulating HSC but questionable for its capacity to generate many of them. I see no problem with that but this makes your reasoning (as discussed in 15) even more problematic. Because of that I also think that putting “robust” in the title of the paper is incorrect.

We again appreciate the Reviewer’s perspective. The Reviewer is indeed correct that very few repopulating units are generated/embryo explant, as our new Limiting Dilution Transplantation data now shows (please see Figure 3A). As requested, we have now removed “robust” from the title of the paper.

27) I think Discussion can be shorter and better structured if you could consolidate your arguments in a more focused and concise text. At the moment it is too diluted.

We have now significantly revised our Discussion.

28) Line 568 – plated in methylcellulose in which concentration?

We apologize for omitting these experimental details in our original submission. Approximately 60,000 sorted cells were dispensed to 3 mL of M3434 and distributed into two plates (#27150, STEMCELL Technologies, Vancouver, Canada). This has been now added to the Methods (line 582-583).

29) Figure 1B shows results of 16 independent experiments according to Legend. Approximately 20 mice were transplanted in total. It is a bit unclear. How many recipients were transplanted in each experiment? Did they transplant yolk sac and embryo bodies from the same embryos in same experiments?

We apologize for any confusion. We have now edited the legend of Figure 1 for clarity.

Specifically: “A) Em and YS were dissected from E8-E8.5 (2-7sp) concepti and cultured as explants at the air-liquid interface for 9-10 days. Explants were then harvested, dissociated and transplanted at \geq four embryo equivalent (e.e.)/recipient. B) CD45.2+ (i.e. 2-3sp Em-Ex (n=7), 5-7sp Em-Ex (n=11), 2-3sp YS-Ex (n=5) or 5-7sp YS-Ex (n=11)) contribution to PB of primary recipients. Data is pooled from 16 independent experiments. In general, tissues from one independent litter were transplanted per experiment. For 2-3sp explants, all YS-Ex (n=5) were transplanted in parallel with Em-Ex isolated from the same concepti. For 5-7sp explants, all transplanted YS-Ex (n=11) were transplanted in parallel with at least one Em-Ex isolated from the same concepti as a positive control for engraftment.”

30) Figure 2A also shows results of repopulation in 5 independent experiments. Three different populations were transplanted into 5 recipients each. Does it mean that every time you transplanted each population just in one recipient? This requires clarification.

We again apologize for any confusion. For each experiment, one recipient was transplanted per sorted population. The figure legend has now been edited for clarity.

31) Fig 5 Legend – “...estimated as %GFP+ amongst the %of tdTomato+ and % of GFP+ cells” - clarification is required.

The figure legend has now been edited for clarity. The %tdTomato+ and %GFP+ cells represents the total Em-Ex-derived PB. The legend now reads “%Lyve1-Cre dependent GFP labeling estimated as %GFP+ amongst % of Explant-derived PB cells is shown”.

Response to Reviewer #3:

The origin of definitive hematopoietic stem cells (dHSCs) has been debated in the past. Currently, most believe that definitive HSCs are from AGM regions. This is largely based on several previous *ex vivo* culture studies that demonstrate definitive HSCs can originate from embryo proper (EM) but not yolk sac (YS). Given that the contribution to the definitive hematopoiesis in those assays were low (1-5%), some in the field still think YS may contribute at levels that are very hard to detect. Using lineage-tracing, it has been reported that Lyve+ cells contribute significantly to dHSCs. But the specificity of Lyve expression may not be carefully examined. In this manuscript, Ganuza et al addressed these specific questions by employing several new techniques, including longer-term *ex vivo* tissue culture and careful examination of Lyve expression. The authors found that EM but not YS are sites where definitive HSCs originate from in *ex vivo* culture. Using longer-term *ex vivo* culture, the authors also demonstrated that Lyve+ cells from EM *ex vivo* culture could generate dHSCs. Overall, the experiments are carefully done and the data are of good quality and convincing. At present, the manuscript emphasizes too much on the technical aspects of a particular paper (ref 14). I agree with the authors' points. Since this field is controversial in the past and is trending to be settled (EM is the origin of dHSCs), I would suggest the authors emphasize more on novel insights (eg single cell expression analysis) and potential experiments in the future to definitively address the question *in vivo*. This will help broaden the interest of the current study.

We are pleased that the Reviewer found our study convincing. We also appreciate the Reviewer's perspective on the focus of the manuscript and have now significantly reorganized it in an effort to place greater emphasis on the major findings of our study and less emphasis on refuting Lee *et al. Cell Reports* 2016.

Below are specific comments:

1. The longer-culture system allowed the authors to achieve more robust reconstitution from *ex vivo* cultures. While I appreciate the amount of work and thoroughness, these experiments are not fundamentally different from the earlier experiments. The conclusion is the same from earlier experiments as well. These experiments measure potential rather than what is really happening *in vivo*. Some discussion on this will be helpful for the field to move forward and provoke more thoughts.

Again, we value the Reviewer's perspective on our study. Although we agree that our approach is similar to previous studies (e.g. Cumano *et al*, 2001), we respectfully submit that our system is a significant step forward in efforts to interrogate the HSC potential of tissues isolated from pre- and peri-circulation concepti. We have established an experimental platform that allows for the emergence of functional HSC with robust engraftment potential from 2-3sp embryo explants. To our knowledge, this has never been previously reported. None-the-less, we fully acknowledge that our *in vitro* system is very likely an imperfect surrogate for the *in vivo* situation – indeed, as are all *in vitro* platforms. We now address this explicitly in our manuscript in lines 359 and 370-372. As requested, we also now include discussion of how information might be gleaned from this experimental platform to identify new useful markers of developing hematopoietic

precursors and inform efforts to direct the fate of pluripotent stem cells (please see lines 216-220; 260-264 and 390-392).

2. It is very interesting that the authors found potential gene signatures for dHSCs in the embryo (Fig. 3). Can any surface antigen from PC9 be used to further purify cells from the culture? What can be done with these genes in the future? Some discussion will really provoke positive thinking and move the field forward.

We are pleased that the Reviewer found our scRNA profiling analysis of Embryo-Explant and YS-Explant-derived cells interesting and valuable. As requested, we have expanded on our discussion of the utility of this dataset in our revised manuscript (please see lines 216-220 and 390-392). Further, based on feedback from Reviewer 2, we have also extended our analysis of this dataset by comparing the transcriptional profiles of these explant-derived populations to existing scRNA datasets of hematopoietic precursors and progenitors isolated from throughout ontogeny (please see Figure 3Di-ii and lines 235-264). Interestingly, the Embryo-Explant-derived ‘X’ population is most similar transcriptionally to E12.5 Fetal Liver-HSC rather than adult HSC (please see Figure 3D). In contrast, YS-explant-derived cells were more similar to more immature hematopoietic precursors. This raises the possibility of exploiting this dataset to identify novel cell surface markers that might enrich for mid-gestation immature HSC from E12.5 fetal liver or differentiating pluripotent stem cells. Indeed, as requested, we have now also interrogated our dataset for cell surface markers unique to population ‘X’ and these results are now presented in Figure 3Dv and Supplemental Figure 3.

3. 1C secondary transplant assay, why not transplanting 3-5sp? I am not asking the authors to perform additional transplantation experiments. But some discussion/clarification will be helpful.

We are happy to address the Reviewer’s inquiry. As 2-3sp embryos are considered pre-circulatory and 5-7sp peri-circulatory, dividing embryos into these two groups seemed rationale (Ji RP *et al.*, *Circ Res* 92, 133-135 (2003); McGrath, K. E. *et al.*, *Blood* 101, 1669-1676, (2003)).

Minor points:

1. Line 97, typo ‘35sp’.

Respectfully, we were indeed referring to published data on E10 embryos. Thus, this was not a typographical error. This study indicated that in E10 embryos (prior to 35sp), the uneven distribution of erythroblasts within the embryo vasculature prior to 35sp suggests that fully functional circulation is established after E10.

2. Line 295, typo ‘Figure 5c’.

We thank the Reviewer and have corrected this to refer to the appropriate Figure.

Response to Reviewer #4:

In this paper the authors use transplantation assays in the murine system to address a central long-standing and largely, but not completely, resolved question in the field of developmental hematopoiesis- the intra- vs. extraembryonic origin of hematopoietic stem cells. As the authors point out, the current consensus is that hematopoietic stem cells arise from large arterial vessels of the embryo proper and not from the yolk sac where the first hematopoietic progenitors and maturing blood cells are first found during embryogenesis. However, a recent report using Lyve1 expression (Lee, et al.) comes to the conclusion that 40% of circulating blood cells in the adult arise from an extraembryonic source. The current submission tackles this Lyve1 publication head on and provides evidence for the current consensus that hematopoietic stem cells arise from intra-embryonic sources. As such, current paradigms are not shifted but this paper provides further evidence for an intra-embryonic source of long-term hematopoiesis and represents a significant contribution to the field.

We are pleased that the Reviewer considers our study an important contribution to the field.

The experiments described in this paper are technically challenging, since they require careful dissection of staged mouse embryos, prolonged explant cultures, and hematopoietic stem cell transplantation studies. This explant culture system was developed in the 1980's and has been used more recently by the Medvinsky laboratory to define pre-hematopoietic stem cell populations in the developing mouse embryo. Importantly, the authors use this explant system to examine mouse embryos just prior to or at the onset of cardiac function and blood cell circulation, that is, prior to significant the intermixing of hematopoietic progenitors and maturing blood cells from the yolk sac into the embryo proper.

The authors show convincingly that long-term hematopoietic stem cell activity emerges from explanted embryo proper and not from explanted yolk sac tissues. They examine the ability of yolk sac and embryo proper tissues explanted for various periods of time to provide stem cell activity. This provides increased confidence of the negative result for yolk sac and they also find that prolonged explant of E9.5 embryo proper leads to a loss of hematopoietic stem cell activity, pointing to a window in HSC emergence.

We very much appreciate the Reviewer's positive comments and have responded to their queries below.

Analyzing the explanted cells, they find that hematopoietic stem cell activity is restricted to the CD31+CD45+Kit+CD41- population, consistent with its emergence from hemogenic endothelium. Why was CD31 rather than VE-cadherin, which has been associated with pre-HSC activity, chosen for this analysis?

We appreciate the Reviewer's question and recognize that VE-cadherin has been a very useful cell surface marker of hemogenic endothelium *in vivo*. VE-cadherin, like CD31, is expressed by endothelium. The CD31+CD45+Kit+ phenotype was chosen because it has

been previously been shown to harbor repopulating activity in embryo explants and thus seemed most apropos to our study (Taoudi et al. Cell Stem Cell 3, 99–108, July 2008).

Single cell analysis identifies a small subset of this CD31+CD45+Kit+ population that differs between embryo proper and yolk sac explants, and the expression of genes associated with hematopoietic stem cells provides indirect evidence that this subset may represent bone fide hematopoietic stem cells. What proportion of the CD31+CD45+Kit+ cell population does this subset constitute?

Are there any cell surface markers associated with this “Z” subpopulation?

We thank the Reviewer for this comment. Based on the Reviewer’s inquiry, we have now strengthened our study by including limiting dilution transplantation analysis of Embryo-Explant-derived cells (please see Figure 3A). We now show that there are only 0.83 repopulating Units (RUs) per cultured Embryo-explant. Thus, the repopulating population is very rare indeed. In the single cell RNA analysis of CD31+CD45+Kit+ cells, 49 concepti were dissected and cultured separately as YS- or Embryo-Explants. Thus, we would expect only about 40.67 RUs (*i.e.* 49*0.83 RUs) in this experiment.

Based on feedback from Reviewer 2, we have also now compared our dataset with transcriptional profiles of embryonic hematopoietic precursors and progenitors throughout ontogeny and identified a subpopulation (designated ‘X’) present only in cultured Embryo-Explants that was transcriptionally quite similar to FL HSC (please see Figure 3Di-ii). This subpopulation was comprised of 623 cells (17.8 % of the Em-Ex population) (see current line 237).

As requested, we have also now identified cell surface markers that could prove useful for enriching “X” cells from cultured Embryo-Explants (please see Figure 3Dv and Supplemental Figure 3).

Given the findings of Lee, et al., the authors go on to examine Lyve1 expression in the developing mouse embryo and find its expression within the pre-hematopoietic stem cells of the embryo proper using the immunophenotypic definitions established by the Medvinsky lab. Importantly, the authors show that Lyve1+ cells derived from embryo proper explants were capable of reconstituting adult recipients. Given the conclusions of the Lee paper, were transportation studies performed with Lyve1+ cells derived from yolk sac explants (even though no HSC activity is present in yolk sac explants)? Was Lyve1 expression found in the “Z” single cell population?

We thank the Reviewer for their inquiry. As requested, we have now performed transplantation studies of YS-explants derived from *Lyve1*^{+Cre}*Rosa26*^{+mTmG} concepti. As expected based on our previous transplantation studies of E8.5 and E9.5 wild type YS-explants (please see Figure 1), we did not observe any peripheral blood repopulating activity from E9.5 *Lyve1*^{+Cre}*Rosa26*^{+mTmG} YS-explants (please see Figure 5D).

As we have updated the analysis of our scRNA data, we have now examined the ‘X’ population for *Lyve1* expression (rather than the previously identified ‘Z’ population).

***Lyve1* did not pass the gene filter in our analysis, meaning it is expressed at very low levels in these cells. Only four CD31+CD45+Kit+CD41- Em-Ex cells and one CD31+CD45+Kit+CD41- Ys-Ex cell displayed detectable *Lyve1* expression at the mRNA level. This is not surprising, given that these cells represent relatively more mature E12.5 FL-like cells and LYVE1 appears to be expressed by earlier hemogenic precursors (Please see Supplemental Figure 5B).**

Overall the paper is well-written. I would suggest less emphasis in the text on the Lee, et al. paper and the pairing down of some repetitive sentences. This would tighten the paper and make the findings better stand on their own merit, as opposed to their derivation as a “reaction” to the conclusions primarily of the Lee, et al. paper.

We thank the Reviewer for their feedback and have made significant edits to our manuscript in an effort to focus the message de-emphasize direct comparisons to Lee *et al.*

Minor points:

1. Given the current evidence that various proportions of many tissue-resident populations in the adult originate from the yolk sac, the statement on line 434 about the potential of the embryo proper to give rise to “all of life-long hematopoiesis” should be toned down.

We thank the Reviewer for pointing this out. We agree and have now edited this sentence to read: “Our study supports a model in which most lifelong hematopoiesis derives from hematopoietic precursors that develop *de novo* in the embryo proper”.

2. There are 19 genes listed in Figure 3ci.

As previously indicated, we have reanalyzed our scRNAseq data.

3. Will the single cell RNA-Seq data be deposited in a public database?

Yes, the Single-cell RNAseq data discussed in this publication have been deposited in NCBI's Gene Expression Omnibus and are accessible through GEO Series accession number GSE GSE110909 (please see Data availability on lines 609-613).

Reviewers' Comments:

Reviewer #1:

Remarks to the Author:

The authors have been exceptionally thorough in addressing my concerns as well as the concerns of the other reviewers. The data presented are clear and convincing in defining an embryonic vs yolk sac origin of dHSCs. Overall, I think the reorganization and tightening of some of the results and conclusions has mostly clarified and strengthened the manuscript, with one exception:

The analysis in Figure 3, though improved by the tSNE analysis, is still somewhat confusing as presented. It is unclear how the scSeq was compared to previous datasets - "assignments" are unclear and methodology should be described. Why wasn't GSEA used? The graph in 3D is particularly confusing - a bar graph would better reflect the nature of the data. In Diii and Div, different colors could be used to underscore differences. At present, plots are not very convincing.

Reviewer #2:

Remarks to the Author:

The authors mostly addressed my questions and significantly improved the manuscript.

1) I would like to comment on the point emphasized in the discussion (lines 370-372). I am not aware of any evidence, which would suggest that mechanisms underlying productive HSC development in vitro are different from the in vivo process. Although it is theoretically possible, making this a strong discussion point in my opinion is inappropriate and against the spirit of this study. They can keep it in but make it in a more passing form. As it is now it unjustifiably draws too much attention and sounds too critical towards their own study.

2) How do the authors explain their result that E10 yolk sac explants produced HSC after 5 days cultures (Fig.1C)? I believe that previously published data from Medvinsky's lab demonstrated the lack of pre-HSC in the yolk sac at this stage. Could the positive result be achieved due to inclusion of extra embryonic vessels in yolk sac explants or differences between mouse strains used or perhaps differences in culture methods? I do not see a problem if the authors did not separate extra embryonic vessels from the yolk sac but this needs to be clearly stated in the paper.

3) More details of their culture method would be helpful to include for those researchers who will wish to use this method, in particular for E8 embryo cultures. Can authors explain how different it is from the method described by Medvinsky's lab? Were membranes washed prior the culture? Were they freely floating or supported by stands and which volume of medium was used? Did they explant a certain number of embryos on a single membrane?

Reviewer #3:

Remarks to the Author:

All of my comments are sufficiently addressed.

Reviewer #4:

Remarks to the Author:

The authors have responded in a thorough and cogent way to the critiques and suggestions of all four reviewers. They have clarified many technical details in the manuscript, performed new transplantation experiments, reanalyzed their single cell data identifying "population X" with an a transcriptional connection to emerging HSCs and an interesting cell surface phenotype, modified the text to emphasize the novel aspects of this manuscript while deemphasize a polemical response to the Lee, et al. paper. While multiple caveats were added to conclusion statements the text, it is clear that their findings support an intra-embryonic source for HSCs, consistent with

findings in the avian system and multiple previous studies in the murine embryo. The modified text reads very nicely and I do not have any further suggestions for the authors.

Response to Reviews (Reviewers are listed below followed by our responses in bold)

Response to Reviewer #1:

The authors have been exceptionally thorough in addressing my concerns as well as the concerns of the other reviewers. The data presented are clear and convincing in defining an embryonic vs yolk sac origin of dHSCs. Overall, I think the reorganization and tightening of some of the results and conclusions has mostly clarified and strengthened the manuscript, with one exception:

The analysis in Figure 3, though improved by the tSNE analysis, is still somewhat confusing as presented. It is unclear how the scSeq was compared to previous datasets - "assignments" are unclear and methodology should be described. Why wasn't GSEA used? The graph in 3D is particularly confusing - a bar graph would better reflect the nature of the data. In Diii and Div, different colors could be used to underscore differences. At present, plots are not very convincing.

We thank the reviewer for their comments and help improving the manuscript. Regarding Figure 3, we have now included further methodological details on how the scRNAseq was compared to previous datasets in the Methods section (please see lines 579-584). We did not employ GSEA because, to the best of our knowledge, there are few comprehensive gene sets available that recapitulate the unique expression profiles of true long-term HSC population. Thus, to avoid the bias inherent in pre-defined gene sets, we instead chose to directly compare the whole expression profiles of our cell population and the reference populations identified by previous studies.

As suggested, the graph in Figure 3Di has now been replaced by a bar graph.

Additionally, Figures 3Diii and 3Div have a new color scale to better highlight differences.

Response to Reviewer #2:

The authors mostly addressed my questions and significantly improved the manuscript.

We are pleased that the Reviewer found the manuscript significantly improved and appreciate their further comments.

1) I would like to comment on the point emphasized in the discussion (lines 370-372). I am not aware of any evidence, which would suggest that mechanisms underlying productive HSC development in vitro are different from the in vivo process. Although it is theoretically possible, making this a strong discussion point in my opinion is inappropriate and against the spirit of this study. They can keep it in but make it in a more passing form. As it is now it unjustifiably draws too much attention and sounds too critical towards their own study.

We thank the Reviewer for their suggestion. We have now removed this sentence from the discussion.

2) How do the authors explain their result that E10 yolk sac explants produced HSC after 5 days cultures (Fig.1C)? I believe that previously published data from Medvinsky's lab demonstrated the lack of pre-HSC in the yolk sac at this stage. Could the positive result be achieved due to inclusion of extra embryonic vessels in yolk sac explants or differences between mouse strains used or perhaps differences in culture methods? I do not see a problem if the authors did not separate extra embryonic vessels from the yolk sac but this needs to be clearly stated in the paper.

We thank the Reviewer for this comment. The reason for including these data in Figure 1C was simply to show that cultured E10.5 and E11.5 Yolk Sac-explants and Embryo-explants were capable of giving rise to HSC under our culture conditions and thereby demonstrate that these conditions are not detrimental to HSC production in general from the Yolk Sac. We do not claim that the HSC precursors in these experiments are autonomous to the Yolk Sac, as here they could readily have migrated from the intraembryonic tissues after the onset of the circulation.

3) More details of their culture method would be helpful to include for those researchers who will wish to use this method, in particular for E8 embryo cultures. Can authors explain how different it is from the method described by Medvinsky's lab? Were membranes washed prior the culture? Were they freely floating or supported by stands and which volume of medium was used? Did they explant a certain number of embryos on a single membrane?

We thank the Reviewer for their comments and have expanded the culture details on the methods section (please see lines 506-507; 516-521).

Reviewer #3 (Remarks to the Author):

All of my comments are sufficiently addressed.

We are very pleased that the Reviewer found their comments sufficiently addressed.

Reviewer #4 (Remarks to the Author):

The authors have responded in a thorough and cogent way to the critiques and suggestions of all four reviewers. They have clarified many technical details in the manuscript, performed new transplantation experiments, reanalyzed their single cell data identifying "population X" with an a transcriptional connection to emerging HSCs and an interesting cell surface phenotype, modified the text to emphasize the novel aspects of this manuscript while deemphasize a polemical response to the Lee, et al. paper. While multiple caveats were added to conclusion statements the text, it is clear that their findings support an intra-embryonic source for HSCs, consistent with findings in the avian system and multiple previous studies in the murine

embryo. The modified text reads very nicely and I do not have any further suggestions for the authors.

We thank the Reviewer for their comments and are delighted to have addressed their suggestions and concerns.

Reviewers' Comments:

Reviewer #1:

Remarks to the Author:

The authors have addressed all of my concerns.

Reviewer #2:

Remarks to the Author:

My comments have been addressed. Thanks.

REVIEWERS' COMMENTS:

Reviewer #1 (Remarks to the Author): The authors have addressed all of my concerns. We thank the Reviewer

Reviewer #2 (Remarks to the Author): My comments have been addressed. Thanks. We thank the Reviewer